# Enhanced surface colonisation and competition during bacterial adaptation to a fungus

Anne Richter[1,2], Felix Blei[2,10], Guohai Hu[1,3,4,5], Jan W. Schwitalla[2],
Carlos N. Lozano-Andrade[1], Jiyu Xie[6], Scott A. Jarmusch[7],
Mario Wibowo[7,11], Bodil Kjeldgaard[1], Surabhi Surabhi[2], Xinming Xu[6],
Theresa Jautzus[2], Christopher B. W. Phippen[7], Olaf Tyc[8,12], Mark Arentshorst[6],
Yue Wang[3,4], Paolina Garbeva[8], Thomas Ostenfeld Larsen[7],
Arthur F. J. Ram[6], Cees A. M. van den Hondel[6], Gergely Maróti[9] &
Ákos T. Kovács[1,2,6] ✉

Bacterial-fungal interactions influence microbial community performance of most ecosystems and elicit specific microbial behaviours, including stimulating specialised metabolite production. Here, we use a co-culture experimental evolution approach to investigate bacterial adaptation to the presence of a fungus, using a simple model of bacterial-fungal interactions encompassing the bacterium *Bacillus subtilis* and the fungus *Aspergillus niger*. We find in one evolving population that *B. subtilis* was selected for enhanced production of the lipopeptide surfactin and accelerated surface spreading ability, leading to inhibition of fungal expansion and acidification of the environment. These phenotypes were explained by specific mutations in the DegS-DegU two-component system. In the presence of surfactin, fungal hyphae exhibited bulging cells with delocalised secretory vesicles possibly provoking an RlmA-dependent cell wall stress. Thus, our results indicate that the presence of the fungus selects for increased surfactin production, which inhibits fungal growth and facilitates the competitive success of the bacterium.

Bacteria and fungi share diverse habitats and consequently, a wide span of interactions is observed between them ranging from mutualism to inhibition. These interactions not only influence the structure and ecology of the respective microbial community but also impact the development and evolution of the interacting species[1,2]. Bacteria and fungi can indirectly affect each other by sensing and responding to diffusible signals such as chemoattractants, quorum-sensing molecules, and volatile substances[3–5]. However, several bacterial-fungal interactions (BFIs) require a close vicinity, and even direct contact between the interacting partners. In certain cases, the bacterium

[1]Bacterial Interactions and Evolution Group, DTU Bioengineering, Technical University of Denmark, Kgs Lyngby, Denmark. [2]Terrestrial Biofilms Group, Institute of Microbiology, Friedrich Schiller University Jena, Jena, Germany. [3]China National GeneBank, BGI-Shenzhen, Shenzhen, China. [4]BGI-Shenzhen, Shenzhen, China. [5]Shenzhen Key Laboratory of Environmental Microbial Genomics and Application, BGI-Shenzhen, Shenzhen, China. [6]Institute of Biology, Leiden University, Leiden, The Netherlands. [7]Natural Product Discovery Group, DTU Bioengineering, Technical University of Denmark, Kgs Lyngby, Denmark. [8]Netherlands Institute of Ecology, Wageningen, The Netherlands. [9]Institute of Plant Biology, Biological Research Centre, Eötvös Loránd Research Network (ELKH), Szeged, Hungary. [10]Present address: Department Pharmaceutical Microbiology, Hans-Knöll-Institute, Friedrich-Schiller-Universität, Jena, Germany. [11]Present address: Singapore Institute of Food and Biotechnology Innovation (SIFBI), Agency for Science, Technology and Research, Singapore, Republic of Singapore. [12]Present address: Department of Internal Medicine I, Goethe University Hospital, Frankfurt, Germany. ✉e-mail: a.t.kovacs@biology.leidenuniv.nl

resides inside the cells of its fungal host[1,5]. Often, a combination of indirect influence and physical contact define a BFI. Various examples of phenotypic adaptation during BFI have been described in the literature, including chemotaxis towards the fungal hyphae[6], induction of secondary metabolites[7], attachment and biofilm matrix production by a bacterium on the hyphae[8], and facilitation of bacterial movement along the mycelia[9,10].

In addition to influencing short-term microbial development, BFIs can also impact the evolution of an organism over longer time scales[11]. For example, a mutualistic relationship between fungus-growing ants, their fungal partner *Leucoagaricus gonglyophorus*, and associated actinobacteria was proposed to have existed for at least 45 million years[1,12]. Antimicrobials produced by actinomycetes protect the fungal gardens by selectively targeting parasitic fungi such as *Escovopis* species suggesting an evolved symbiosis[7,13,14]. Endophytic bacteria have been also suggested to genetically adapt in the fungal host, although broad host range bacterial species has been also suggested[15]. Therefore, BFI could serve as a model to study the evolution of novel eukaryote-prokaryote interactions[2].

The rapid growth of microorganisms facilitates experimental evolution studies in the laboratory[16]. However, only few studies have been published that dissect the evolution of ecological interactions in a BFI setup. The evolution of *Staphylococci* in the presence of various fungal partners highlighted changes in secondary metabolite production and biofilm formation[17]. Based on the BFI literature describing a plethora of phenotypic responses of both bacteria and fungi during their interaction, we hypothesized that bacterial adaptation in the presence of a fungus could potentially lead to changes in relevant traits of a bacterium.

A direct interaction between the black mould-causing fungus *A. niger* and the plant growth-promoting Gram-positive bacterium *Bacillus subtilis* has previously been described[8,18]. Both organisms are commonly found in soil, thus they potentially coexist in the same habitat and influence each other, however, their ecological interaction is not scrutinized yet in natural settings. Nevertheless, the examination of two model organisms in the laboratory could reveal new mechanistic insights on BFIs. For example, attachment of *B. subtilis* cells to fungal hyphae results in transcriptional changes. Specifically, the production of antimicrobial substances is downregulated in both microorganisms, including the *B. subtilis*-produced lipopeptide, surfactin. Furthermore, genes related to motility and aerobic respiration are also downregulated in the bacterium during attachment[18].

*B. subtilis* has been exploited for experimental evolution during biofilm formation to dissect the influence of biofilm matrix sharing[19–23] and has been also studied during adaptation to the plant root surface[24–27]. These studies highlighted the rapid genetic adaptation of *B. subtilis* and corresponding phenotypic changes related to bacterial differentiation. Moreover, these works were also facilitated by the extensive knowledge on the molecular details underlying *B. subtilis* physiology, gene regulation, phenotypic differentiation, and secondary metabolite production[28–32]. Therefore, *B. subtilis* offers a fascinating model to investigate the effects of long-term cultivation of this bacterium in the presence of a fungus and to focus on the bacterial genetic and phenotypic evolution.

Here, we used a laboratory adaptation experiment, *B. subtilis* cells with enhanced surfactin production and spreading behaviour are selected, which is mimicked by incorporation of specific mutations in genes encoding a global two-component system. Increased surfactin production and niche colonisation by the bacterium disrupt fungal expansion and acidification of the medium. Further, we discover that surfactin causes mislocalization of secretory vesicles in *A. niger*, indirectly inducing cell wall stress in the fungus.

## Results

### Adaptation of *B. subtilis* to the presence of *A. niger*

Utilising the ability of *B. subtilis* to colonise and grow on mycelia of the fungus *A. niger*[8,18], a simple bacterial-fungal co-culture system was established in which fungal spores and subsequently diluted planktonic bacterial cultures were streaked onto agar medium in a hashtag and square pattern. Areas where bacteria grew over fungal hyphae were dissected (Fig. 1a). This agar block was used to inoculate the bacterium on fresh medium, followed by planktonic growth for 24 h before initiating a new co-culture cycle. Importantly, *A. niger* was not evolved in this setup; a fresh batch of spores was used each time from the same fungal stock. Five parallel co-cultivated evolution BFIs (denoted CoEvo) were used in addition to five control lineages of the bacterium cultivated alone (denoted Bac) following otherwise identical isolation procedures (Supplementary Fig. 1a). After 10 weekly transfers, two endpoint isolates were collected from each lineage of both setups. These evolution endpoint bacterial isolates were tested for growth in the presence of the fungus and their ability to form a colony biofilm. Data for isolates 1 from each lineage are included in the figures and datasets describing the phenotypic assays, as the second isolates behaved comparable to the first isolate from the same lineage, except Bac3 evolved isolates during colony biofilm development.

First, bacterial cultures were spotted between two lines of 1-day-grown *A. niger* mycelia. The ancestor bacterium grew and created a small colony almost surrounded by the fungus after 7 days (Fig. 1b). Most of the evolved isolates behaved similarly to the ancestor, but isolates from two CoEvo lineages, especially CoEvo2 and partially CoEvo4, displayed increased spreading that limited fungal growth and expansion (Fig. 1b and Supplementary Movie 1 and 2). All Bac isolates behaved like the ancestor. In addition to CoEvo2 and CoEvo4, CoEvo3 also displayed increased surface spreading in the absence of the fungus unlike in the presence of *A. niger* (Supplementary Note 1, Supplementary Fig. 2). Preliminary experiments suggest that fungal produced volatile compounds, dimethyl disulphide, pyrazine, and 1-pentyne restrict CoEvo3 spreading (Supplementary Note 1, Supplementary Dataset 2, Supplementary Fig. 2).

To assess the influence of bacterial spreading on fungal physiology, bacterial cultures were spotted next to a 1-day-grown *A. niger* streak on lysogeny broth (LB) plates containing the pH dye Bromocresol Purple that is purple above pH 6.8 and yellow bellow pH 5.2. When the fungus was cultivated alone, large section of the agar medium (>30%) displayed a yellow colour corresponding to the ability of *A. niger* to acidify its environment via excretion of organic acids[33,34]. The ancestor bacterium, Bac and most of the CoEvo isolates restricted acidification and the yellow coloration was apparent on the edge of the fungal streak, while the CoEvo2 isolate almost fully supressed the acidified area as quantitatively determined by measuring the yellow area of the plate (Fig. 1c and Supplementary Fig. 3a, b). The area of acidification was increased when *srfAC* mutant *B. subtilis* was tested (*srfAC* strain unable to produce the antimicrobial lipopeptide surfactin[32], see below), displaying a relative acidification between the uninoculated control and the other *B. subtilis* strains (Supplementary Fig. 3b). These results suggest increased surface colonisation by the CoEvo2 lineage in the presence of the fungus that restrict the fungus-mediated acidification, whereas such adaptation wasn't observed for replicates of the Bac and other CoEvo isolates. Notably, the surface colonization of the ancestor *B. subtilis* was not increased when an oxaloacetate hydrolase encoding *oahA* mutant *A. niger* strain was used that lacks medium acidification, while the CoEvo2 colony phenotype was similarly increased in the presence of *oahA* mutant *A. niger* as with the wild-type fungus (Supplementary Fig. 3c), suggesting that the evolution of CoEvo2 was not primarily driven by increased acid resistance.

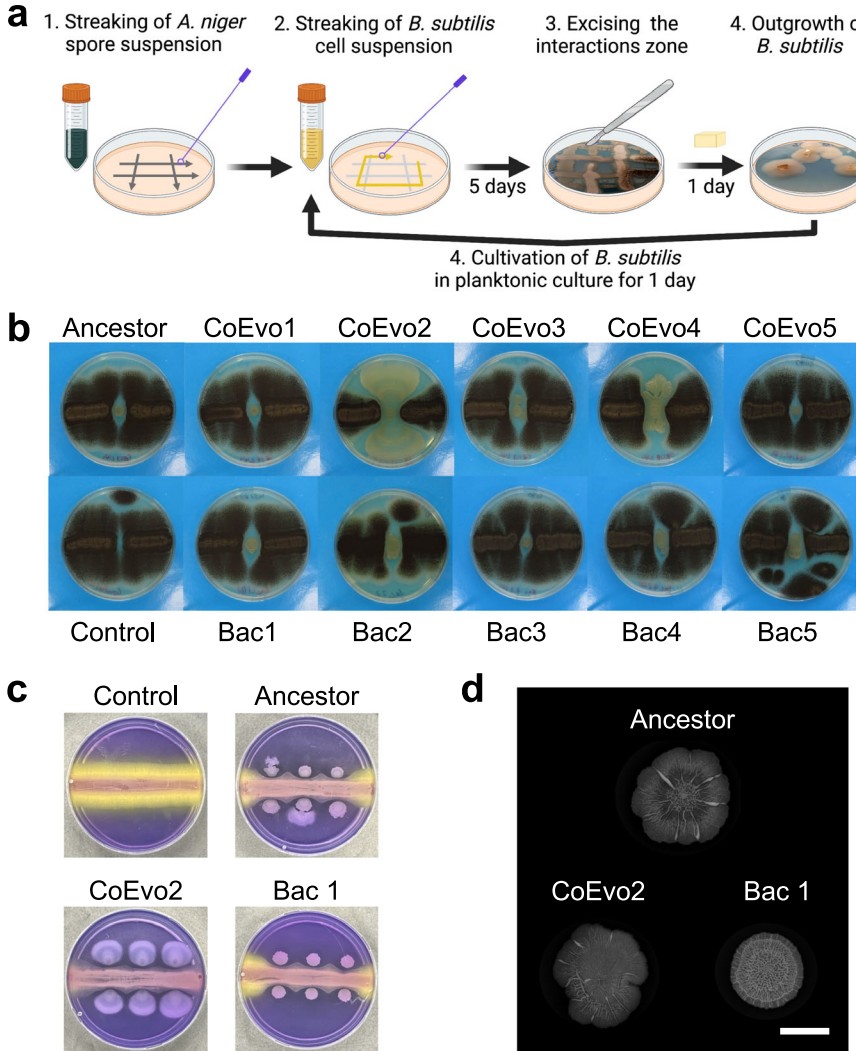

**Fig. 1 | *B. subtilis* adaptation to the presence of *A. niger.* a** Schematic representation of the evolution setup. Panel (**a**) was created with BioRender.com, released under a Creative Commons Attribution-NonCommercial-NoDerivs 4.0 International license. **b** Bacterial growth spotted between two lines of fungal streaks. **c** Panels showing the pH of the medium using Bromocresol Purple, where bacterial cultures were spotted next to a continuous fungal spore streak. Purple and yellow colours indicate pH > 6.8 and pH < 5.2, respectively. In **b** and **c**, the plate size = 9 cm. CoEvo refers to co-culture evolved isolates, Bac denotes bacteria only evolved isolated. **d** Biofilm colonies of selected evolved isolates. Scale bar = 10 mm. All isolates for panel (**c**) and (**d**) are shown in Supplementary Fig. 3. Experiment was repeated 3 times with comparable colony biofilm structures.

## Evolution on agar medium selects for wrinkled variants

Unlike in the presence of a fungus, evolution of *B. subtilis* on a solid agar surface as pure cultures could potentially be driven by adapting biofilm complexity. Indeed, all but one replicate (Bac 3.2) of Bac isolates displayed increased architecture complexity (Fig. 1d and Supplementary Fig. 3d) reminiscent of biofilm matrix-overproducing wrinkled colonies observed in previous experimental evolution settings[22,24,25,35]. In general, isolates from CoEvo lineages exhibited similar colony structure to the ancestor, with a slight increase in complexity of evolved isolate CoEvo4.1 (Supplementary Fig. 3d).

## Mutation of DegU enhances spreading

To dissect the mutational landscapes of *B. subtilis* adapted in the presence or absence of *A. niger*, sequential populations of each lineage at each transfer were subjected to high-coverage metagenome sequencing (except the first three timepoints of four Bac lineages). Single-nucleotide polymorphisms (SNPs) and short insertions and deletions (indels) were determined using the *breseq* pipeline[26,36,37] which detected 142 and 157 mutations with >5% frequency in CoEvo and Bac populations, respectively (frequencies of each mutated gene

in each lineage are shown in Supplementary Fig. 4a and the mutation types are shown in Supplementary Dataset 1). Subsequently, the genealogical structure of each lineage was determined and visualised as lineage frequencies from shared, nested mutation trajectories over time, using *Lolipop*[38,39]. This approach revealed both unique dominant mutations, including SNPs in *degU* and *degS* genes in lineages CoEvo2 and CoEvo3, respectively, and parallel genetic changes, such as specific SNPs in *sinR* of Bac lineages (Fig. 2a and Supplementary Dataset 1). In addition, certain genes were commonly mutated under both conditions, including *ywdH* and *opp* genes. Importantly, while genetic diversity was comparable in both CoEvo and Bac lineages (Fig. 2b), parallelism was higher in control evolved lineages than fungal-adapted lines (Fig. 2c), based on the Jaccard index that reflects the likelihood that the same gene is mutated in two independent lineages[40].

Subsequently, mutations in the genomes of the single endpoint isolates from each lineage were determined to corroborate population sequencing (Supplementary Dataset 1). CoEvo- and Bac-specific mutations were identified in addition to the few commonly mutated genes (Fig. 2d). SNPs in *sinR* were identified in Bac-evolved lineages that were previously associated with increased colony

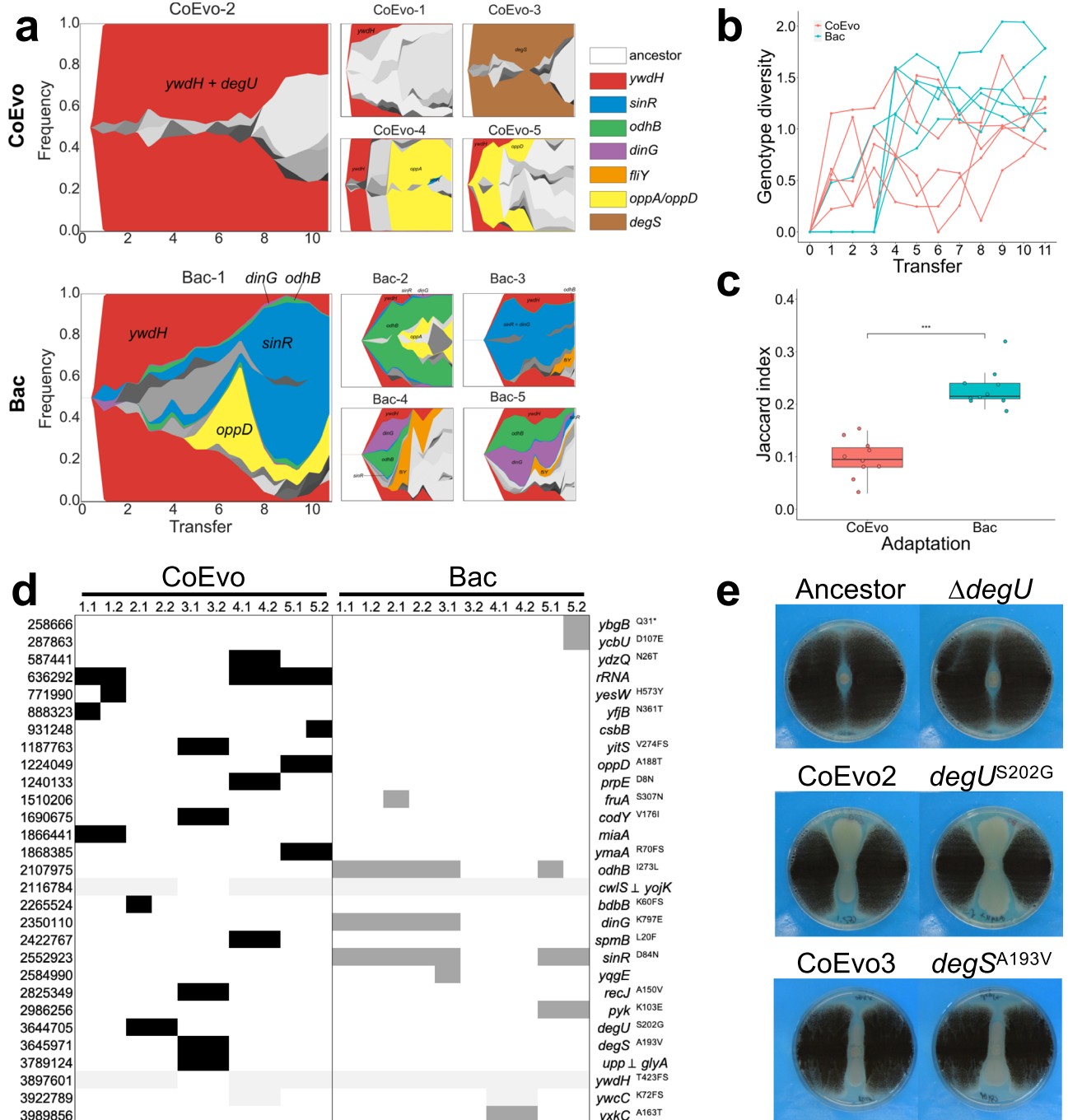

**Fig. 2 | Genetic characterisation of *B. subtilis* adaptation to *A. niger*. a** Graphs showing the genealogy and genotype frequencies throughout the transfers. Each colour or shade represents a distinct genotype. The vertical area corresponds to genotype frequency as inferred using Lolipop. Dominant genotypes with a high mutation gene frequency, which are shared in different populations, are highlighted by matching colours within both adaptation models. Grey colour refers to nested genotypes. **b** Genotype diversity. The dynamic distribution of genotype alpha diversity in each population of the two adaptation models over time was calculated using the Shannon method. **c** Degree of parallelism within both conditions estimated by Jaccard index. Asterisks denote significant differences (***$p = 1.698\ 10^{-7}$, Student's unpaired two-tailed *t*-test, $n = 10$). Boxes represent the first and the third quartile, lines indicate the median, the bars span from the maximum to the minimum value, and dots indicate the J value of each condition. **d** Detected mutations in endpoint CoEvo and Bac isolates. Black indicates CoEvo-specific mutations, dark grey denotes Bac-specific SNPs and light grey represents common mutations found in both experimental setups. ⊥ indicates intergenic regions between the two genes depicted. * indicates an introduced stop codon. Amino acid changes are indicated unless synonymous mutations were determined. **e** Bacterial colony growth between two lines of fungal streaks, including the ancestor, CoEvo isolates, and mutants (*degU* deletion, and *degU*^S202G or *degS*^A193V point mutations introduced to the ancestor). Plate size = 9 cm.

wrinkling[24,35,41,42]. Interestingly, isolate Bac 3.2 lacking increased colony wrinkles harboured no unique SNPs other than those also present in CoEvo isolates. Isolates from lineages CoEvo2 and 3 possessed SNPs in the genes coding for the two-component regulatory system, DegU and

DegS, respectively. The serine to glycine substitution at position 202 of DegU is located at the edge of the helix-turn-helix domain involved in DNA binding (Supplementary Fig. 4b). Mutations in the vicinity (e.g., DegU^R184C in ref. [43] and DegU^II86M and DegU^H200Y in ref.[44]) have been

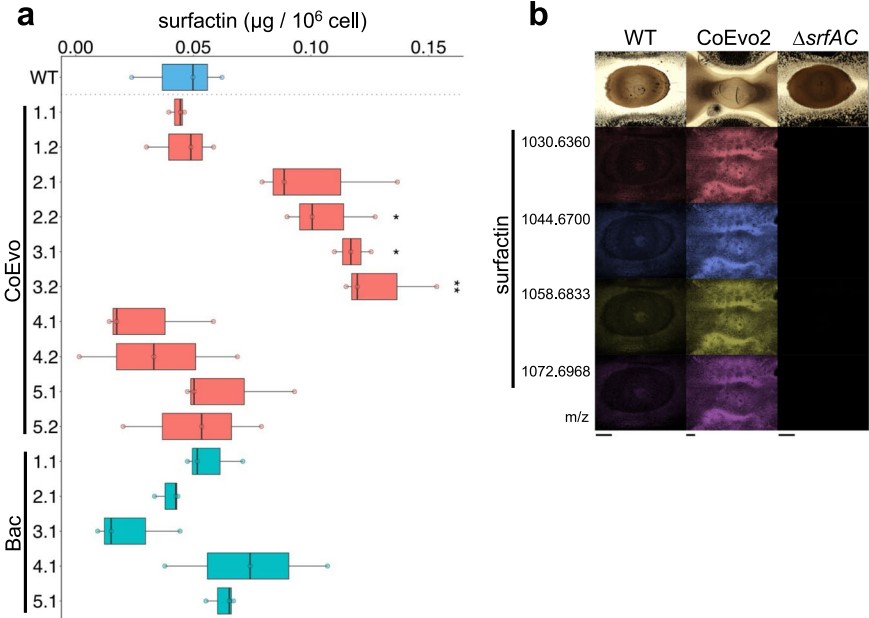

**Fig. 3 | Detection of surfactin in evolved bacterial strains. a** UHPLC-HRMS quantification of the produced surfactin normalised against the number of cells in colonies on agar medium. A two-sided Student's *t*-test with Bonferroni–Holm correction was performed using WT strain as reference (*$p_{adjust}$ WT vs. CoEvo2.2 = 0.018, *$p_{adjust}$ WT vs. CoEvo3.1 = 0.0139, **$p_{adjust}$ WT vs. CoEvo3.2 = 0.0072, in all cases *n* = 3). The black centre line denotes the median value of surfactin production normalized by cells in colonies (50th percentile), while the boxes contain the 25th to 75th percentiles of the dataset. The whiskers mark the 5th and 95th percentiles. The overlays dots are the raw measure of surfactin of each replicate. **b** MALDI-MSI spatial detection of surfactin isoforms in bacterial colonies (wild-type, CoEvo2, and the *srfAC* mutant from left to right) grown between two fungal streak lines. m/z values of surfactin isoforms are indicated on the left. Scale bars = 2 mm. MALDI-MSI experiment was performed twice for replication of the results.

reported to alter the expression of DegU-regulated genes. The alanine to valine substitution at position 193 of DegS (Supplementary Fig. 4c) has been previously characterised and is known to abolish DegS protein kinase activity[45]. Reduced phosphorylation of DegU by DegS results in upregulation of genes related to motility and natural competence for DNA uptake, whereas genes encoding secreted degradative enzymes are upregulated by high levels of phosphorylated DegU[43,45].

When *degU*[S202G] and *degS*[A193V] mutations were reintroduced into the ancestor, the engineered strains phenocopied the evolved isolates in the presence of *A. niger* (Fig. 2e, Supplementary Fig. 2 and 3a–c, and Supplementary Note 1). By contrast, deletion of *degU* was comparable to the wild-type ancestor (Fig. 2e), indicating that these mutations modified but did not abolish DegU activity.

Building upon a previously developed quantitative fungal inhibition test[46], we demonstrated that the growth of *A. niger* was similarly reduced by CoEvo2 and *degU*[S202G] strains compared with the ancestor *B. subtilis* (Supplementary Fig. 4d).

### Surfactin production is enhanced in evolved isolates

Undomesticated *B. subtilis* isolates produce a set of non-ribosomal peptides, including the two lipopeptide surfactin and plipastatin involved in fungal inhibition[32]. The lipopeptide surfactin plays an important role in *B. subtilis* expansion over semi-solid surfaces including swarming or sliding[47–50], and genes encoding the surfactin biosynthesis apparatus were downregulated during the initial attachment of *B. subtilis* to hyphae of *A. niger* in planktonic cultures[18]. Therefore, we hypothesised that surfactin production might be altered in CoEvo2 after adaptation to the presence of the fungus. Direct semi-quantification of surfactin produced by the bacterium grown on agar medium confirmed an increase in surfactin production for CoEvo2 and 3 isolates compared with the ancestor (Fig. 3a, Student's *t*-test with Bonferroni-Holm correction), possibly facilitating the improved expansion of the bacterium on the agar surface. By contrast, surfactin

production by Bac strains was not increased (Fig. 3a, Student's *t*-test with Bonferroni-Holm correction). Drop collapse assay showed that *degU*[S202G] and *degS*[A193V] alleles recreate the increased surfactant properties observed in CoEvo2 and CoEvo3 strains, respectively (Supplementary Fig. 5a). Matrix-assisted laser desorption/ionisation-mass spectrometry imaging (MALDI-MSI) analysis of co-inoculated bacteria-fungi samples further demonstrated increased surfactin production by CoEvo2 compared with the ancestor strain (Fig. 3b). By contrast, production the lipopeptide plipastatin by CoEvo2 was comparable with the ancestor (Supplementary Fig. 5b).

### Surfactin causes hyphal bulging and cell wall stress

The fungus-adapted bacterial isolate CoEvo2 displayed increased surfactin production and surface colonisation. In addition, acidification of the environment by *A. niger* was reduced in the presence of CoEvo2, while enhanced in the presence of *srfAC* mutant. Therefore, we hypothesised that increased surfactin production not only stimulated spreading of CoEvo2 colonies, but also influenced the fungus. Intriguingly, when planktonic cultures of *A. niger* were supplemented with bacterial cell-free supernatant from the *B. subtilis* ancestor, some fungal hyphae displayed bulging (Fig. 4a). However, bulbous fungal cells were absent when surfactin was not produced in the bacterial cultures due to deletion of either *srfAC* (encoding a subunit of the surfactin biosynthesis machinery) or *sfp* (encoding a phosphopantetheinyl transferase involved in activation of the peptidyl carrier protein domains of non-ribosomal synthetases; Fig. 4a). Supplementation the fungal culture with pure surfactin was sufficient to induce bulging (Fig. 4a), and bulbous hyphal cells were still present in mutants lacking the ability to synthesise other non-ribosomal proteins (those that are involved in the synthesis of plipastatin and bacillaene) unless surfactin biosynthesis was simultaneously disrupted (Supplementary Fig. 6a).

To investigate the influence of surfactin on fungal hyphae, we tested a series of fluorescent reporter strains for which specific cell

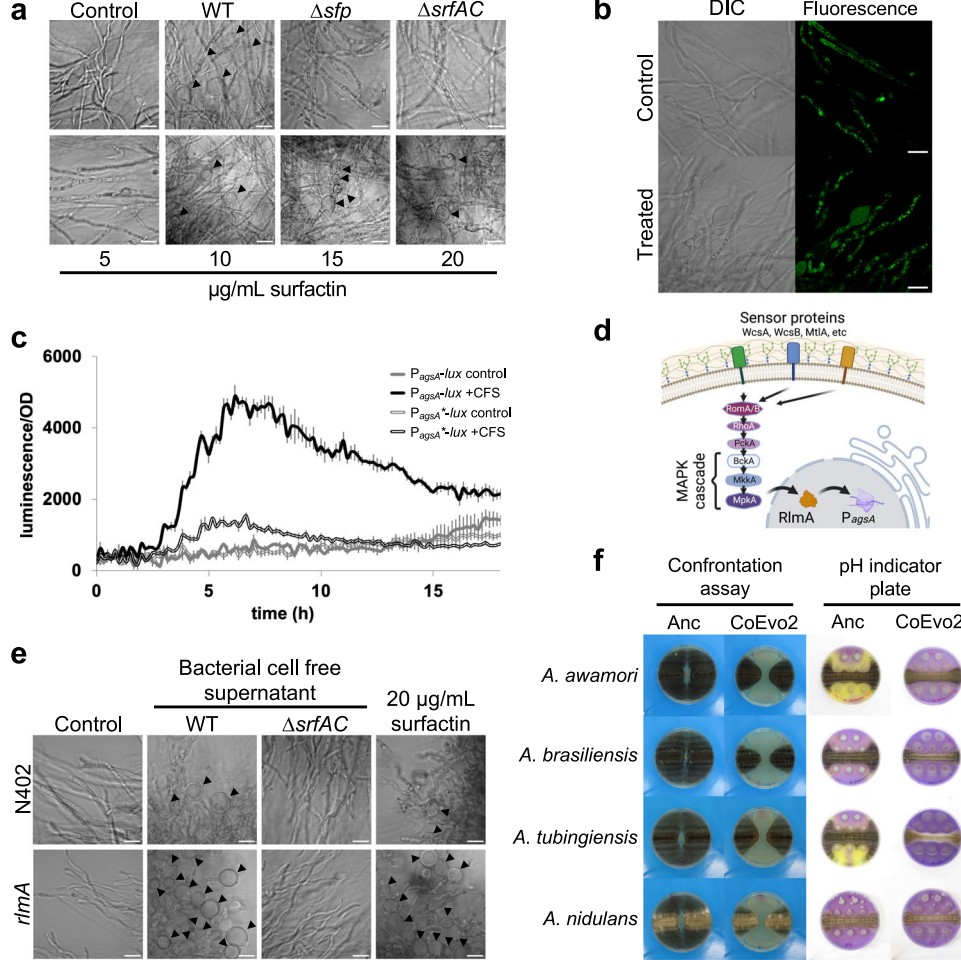

**Fig. 4 | Influence of surfactin on fungal hyphae. a** Microscopy visualisation of bulging fungal hyphae from shaken cultures with cell-free supernatant (CFS) of wild type (WT), *sfp* and surfactin mutants (top panels) or increasing amounts of surfactin (bottom panels). Scale bar = 20 μm. Experiment was performed three times. **b** DIC (left) and green fluorescence (right) imaging of the *A. niger* FG7 strain (P$_{synA}$-GFP::SynA). No autofluorescence was detectable with the CLSM settings used. Scale bar = 20 μm. Experiment was performed three times. **c** Luminescence reporter assay using *A. niger* strains MA297.3 and MA584.2 containing P$_{agsA(3×RlmA\ box)}$ and P$_{agsA(RlmA\ box\ mutated)}$ before the promoter-less luciferase (grey and black lines, respectively). Cultures were treated with LB medium (double lines) or cell-free supernatant (CFS, filled lines). Error bars indicate standard deviations. *n* = 3 **d** Schematic representation of cell wall stress perception in *A. niger* according to a previous report[86]. Panel d was created with BioRender.com, released under a Creative Commons Attribution-NonCommercial-NoDerivs 4.0 International

license. **e** Influence of the cell-free supernatant from the WT strain and the surfactin mutant (Δ*srfAC*), and 20 μg/ml surfactin on bulbous hyphae formation in WT and *rlmA* mutant *A. niger* strains (top and bottom panels, respectively). Scale bar = 20 μm. Experiment was performed three times. **f** Impact of the ancestor (Anc) and Coevo2 strains on Aspergillus species *A. awamori*, *A. brasiliensis*, *A. tubingiensis*, and *A. nidulans*. The two columns on the left show bacterial growth spotted between two lines of fungal streaks, while the two columns on the right depict the pH of the medium measured using Bromocresol Purple, where bacterial cultures were spotted next to a continuous fungal spore streak. Two rows of bacterial strain were spotted on the top of the Bromocresol Purple-containing medium agar plates, to observe the influence of increased bacterial biomass on acidification (i.e., ancestor can reduce acidification only when this additional two colonies are added). Purple and yellow colours indicate pH > 6.8 and pH < 5.2, respectively. Plate diameter = 9 cm.

components could be monitored. In the presence of bacterial cell-free supernatant the cell wall, mitochondria, and nuclei were comparable to those of untreated samples, while membranous components including vacuoles, Golgi, and endoplasmic reticulum were aberrant (Supplementary Fig. 6b). In particular, secretory vesicle-specific soluble NSF attachment protein receptor SncA fluorescent reporter showed a homogenous distribution in bulged cells, and the secretory vesicles were mislocalised (Fig. 4b), unlike in untreated samples where these secretory vesicles are located close to the tips of hyphae to deliver cell wall components and secrete fungal enzymes[51]. Further experiments are required to determine whether the SncA-GFP reporter is mislocalized in the bulbous cells or the GFP is released from the SncA protein.

The surfactin-induced bulging of *A. niger* hyphae resembles the impact of Calcofluor White (CFW), a chitin and cellulose-binding fluorescent dye, on fungal mycelia[52]. CFW provokes cell wall stress in *A.*

*niger* including upregulation of α-1,3-glucane synthase encoded by the *agsA* gene[52]. We hypothesized that potentially reduced cell wall synthesis due to mislocalization of secretory vesicles in the presence of surfactin could result in cell wall stress in *A. niger* like that observed in the presence of CFW. Indeed, *agsA* transcription was induced in *A. niger* when treated with bacterial cell-free supernatant when monitoring using luciferase (Fig. 4c) or a fluorescence reporter (Supplementary Fig. 6c). Furthermore, induction was dependent on production of surfactin by *B. subtilis*, as it could also be promoted by supplementation of pure surfactin (Supplementary Fig. 6c and e). The cell wall stress-sensing pathway in *A. niger* includes the transcription factor RlmA that directly activates *agsA* transcription (Fig. 4d). Accordingly, deletion of the RlmA-binding box within the promoter region of the *agsA* reporter construct tempered induction by bacterial cell-free supernatant (Fig. 4c). Finally, disruption of *rlmA* amplified the number of bulbous hyphal cells in the presence of bacterial cell-free

supernatant or purified surfactin (Fig. 4e). As expected, without surfactin in the cell-free culture supernatant, neither wild-type nor *rlmA* mutant *A. niger* hyphae displayed bulging (Fig. 4e).

These results demonstrate that the mode of action of *B. subtilis*-produced surfactin on *A. niger* involves mislocalization of secretory vesicles and hyphal bulging. As secretory vesicles deliver the components required for cell wall synthesis at the tip of the hyphae[53], the mislocalization of the secretory vesicle and potentially reduced cell wall synthesis might inducing cell wall stress. Plipastatin, another lipopeptide synthesised by *B. subtilis*, did not provoke such changes in hyphal cell morphology in *A. niger* (Supplementary Fig. 6a). Inhibition of *Fusarium* (another filamentous fungus) by *B. subtilis* is mediated by plipastatin but unaffected by deletion of surfactin[32]. Therefore, potential bulging of *Fusarium* hyphae has been monitored and tested using *B. subtilis* strains lacking either surfactin or plipastatin. In line with the inhibitory activity of plipastatin against *Fusarium*, bulbous cell formation by *Fusarium culmorum* and *Fusarium oxysporum* was dependent on plipastatin in the cell-free supernatant (Supplementary Fig. 7).

### Spreading-mediated inhibition of other *Aspergillus* species

To reveal whether the ability of fungus-adapted *B. subtilis* to decrease *A. niger* growth was transferable to other *Aspergillus* species, the CoEvo2 isolate was tested on them (Fig. 4f). In addition to *A. niger*, spreading enhanced by the CoEvo2 strain was able to restrict the expansion of *Aspergillus awamori*, *Aspergillus brasiliensis*, *Aspergillus tubingiensis* and *A. nidulans*. Co-inoculation on medium containing a pH indicator revealed that CoEvo isolates could prevent fungal-mediated acidification of the medium during co-cultivation with several *Aspergillus species* (Fig. 4f), demonstrating a general advantage of increased spreading by *B. subtilis* during competition for space and nutrients on a surface.

## Discussion

During prolonged co-cultivation with *A. niger*, a specific lineage of *B. subtilis* was adapted by increasing their surface spreading ability, which allows the bacterium to cover a larger area, reach more nutrients, and thus successfully compete against the fungus. The evolution of such elevated mobility might have been facilitated by the microhabitat generated by the fungal mycelium network. Zhang and colleagues observed rapid movement of single bacterial cells and subsequently groups of cells along a water film around fungal hyphae[54]. The area surrounding a fungal hypha retains water and has a higher humidity than the wider environment, thereby generating conditions that promote bacterial movement even in a dry milieu[54,55]. Spreading along a hypha might be advantageous for *B. subtilis* as the environment dehydrated over the 5-day incubation. Lack of a fungal microhabitat in control experiments may create drier conditions. Indeed, *B. subtilis* generally adapted by increasing matrix production in the absence of fungal mycelia. Investment in biofilm formation and smaller colony size might also be related to the dry environment since biofilms can act as water reservoirs[56,57].

Here, *B. subtilis* adaptation to the presence of *A. niger* in CoEvo2 and 3 lineages also increased competition, since in addition to enhanced spreading, several adapted lineages also produced more surfactin. Secondary metabolites such as lipopeptides are often involved in BFIs. These may act as cues to elicit a specific reaction or interfere directly with another microorganism via chemical warfare[7,58]. While the higher surfactin level of the adapted isolates from CoEvo2 lineage likely increased the spreading behaviour[48,50], it also directly contributed to fungal inhibition via its antimicrobial properties[59,60]. Importantly, we also revealed that the mode of action of surfactin on *A. niger* involves influencing membranous fungal cell components, mislocalization of secretory vesicles that deliver components for cell wall synthesis, and indirectly provoking cell wall stress. By contrast, rapid

adaptation of *B. subtilis* in a 15-day co-culture with the fungus *Setophoma terrestris* was accompanied by reduced surfactin production, increased biofilm development, reduced swarming, and increased emission of anti-fungal volatile compounds[61]. The distinct adaptation route could be potentially explained by differences in fungal partner sensitivity to antimicrobial compounds.

The fate of a BFI from mutualism and co-existence to competition and elimination of their companion is often not only determined by the specific influence of members on each other, but also influenced by their environment. For example, spatiotemporal organisation is crucial for a stable co-culture and the beneficial influence of *B. subtilis* on the fungus *Serendipita indica*[62]. Interactions can also be influenced by environmental parameters including pH[61]. *A. niger* acidifies its environment by organic acid secretion, including oxalic acid[33,34]. One potential bacterial strategy to cope with a fungus-constructed niche includes adaptation to growth at low pH, thereby promoting survival in acidic soil, and hence co-occurrence with various fungal species, as reported previously[63,64]. In our experiment, instead of adapting to a lower pH milieu, the evolved *B. subtilis* isolates possibly prevented medium acidification by *A. niger* rather than actively inhibiting the process. The increased spreading of *B. subtilis* could potentially lower nutrient availability for *A. niger*, surfactin could also inhibit the fungus, and a combination of both mechanisms could contribute to reduced growth and acidification by *A. niger*. This observation is further supported by the comparable influence of the CoEvo2 evolved isolate on a set of *Aspergillus* species.

Genome re-sequencing of the evolved isolates and populations revealed mutations in genes encoding global regulators, including *degU* and *degS* in the fungal co-cultured CoEvo 2 and 3 lineages, respectively, and *sinR* in all Bac lineages evolved on agar medium. While mutations in the *sinR* gene have been identified in numerous studies before[24–27,35,41,42], *degU* has been reported as target for mutation during domestication[44]. Interestingly, increased surfactin production has been detected in the CoEvo2 and 3 isolates; nevertheless, *srfA* operon that encodes the surfactin synthase complex has not been reported to be part of the DegU regulon[65,66]. Further experiments will be required to dissect whether the observed mutations in *degU* and *degS* genes contribute to increased surfactin production through enhancing transcription of *srfA* operon or other genes contributing to surfactin synthesis. Population metagenomics verified the abundance of SNPs in *degU*, *degS*, and *sinR* genes, in addition to confirming that adaptation in a biotic environment (interaction with a fungus) results in lower parallelism compared with an abiotic environment (cultivation on the agar medium without the fungus). This confirms previous observation on experimental evolution of *B. subtilis* and *Bacillus thuringiensis* in abiotic and biotic biofilms[26].

Although the ecological relevance of *Bacillus* and *Aspergillus* interaction and their co-evolution in nature is yet to be demonstrated, previous studies have described the presence or the isolation of *B. subtilis* next to fungi[32,67]. In addition, the level of surfactin production was reported to be variable depending on the isolates[32,68]. Future experiments needed to dissect the prevalence of specific SNPs that we observed in the regulatory genes of evolved populations and its correlation to the fungal abundance in the specific environment.

Overall, our study demonstrates the potential of combining co-culture and laboratory evolution experiments to deepen our understanding of BFIs. Such co-culture adaptation methodology could offer a general, genetically modified organism-free approach to enhance antifungal activities of biocontrol bacteria against pathogenic fungi.

## Methods
### Media and cultivation conditions
*B. subtilis* DK1042 (a naturally competent derivative of the undomesticated biofilm-proficient NCIB3610 strain) and its derivatives were generally grown in lysogeny broth (LB; Lenox, Carl Roth, Germany)

from a frozen stock for 18–20 h at 37 °C with shaking at 220 rpm. To harvest conidia, *Aspergillus* strains were grown on malt extract agar (MEA) plates (Carl Roth, Germany) at 30 °C for 1–2 weeks or on Complete Medium agar plates at 28 °C for 3 days. After incubation, 10–20 ml of sterile Saline Tween solution (8 g/l NaCl with 0.005% Tween 80) was added, conidia were scraped off, and conidia-containing liquid was collected. The solution was vortexed, sterile filtered with Miracloth rewetted funnels and stored at 4 °C. Interactions of *B. subtilis* and *A. niger* on plates were tested using LB medium supplemented with either 1% or 1.5% agar. To follow pH changes, LB agar medium was supplemented with 0.02 g/l Bromocresol Purple. All experiments have been performed using at least 3 independent replicates, and randomly selected representative image is shown in the figures. Inhibition of *A. niger* by *B. subtilis* was quantitated motivated by a previously established method[46]. Briefly, 15 µl spores of *A. niger* AR19.1 ($3 \times 10^6$) harbouring constitutively expressed GFP were inoculated to each well of a 48-well plate in the presence of bacterial cultures with different optical density at 600 nm. The cultures were incubated at 28 °C for 3 days, and the GFP was recorded at 5 positions in each well. Interactions of *B. subtilis* with different fungi in liquid cultures were tested in LB medium supplemented with 10 mM $MgSO_4$ and 1 mM $MnCl_2$ as previously reported[18]. Briefly, fungal spores were inoculated in LB medium and cultivated for 24 h. Subsequently, 1 ml cultures of the five fungal microcolonies were supplemented with 10 mM $MgSO_4$ and 1 mM $MnCl_2$ and overnight-grown *B. subtilis* was added at 1000-fold dilution in 24-well plates. Fungal cell morphology was assessed after 24 h of cultivation at 28 °C with shaking at 120 rpm.

For luminescence reporter measurement, $7.5 \times 10^4$ fungal spores/ml were inoculated in 200 µl LB medium containing 50 µl luciferin (final concentration 0.5 mM) before either 50 µl bacterial cell-free supernatant, CFW (final concentration 50 µg/ml), LB medium or 1% methanol was added.

## Strain construction

*B. subtilis* cells were transformed with genomic DNA (extracted from 168 *degU*) and selected on LB plates supplemented with kanamycin (10 µg/ml) to obtain the *degU* deletion mutant in the DK1042 background. Natural competence was used to transform *B. subtilis*[69]. To obtain the single nucleotide-exchanged strain, DK1042 was first transformed with plasmids pTB693 and pTB694 (for *degU*[S202G] and *degS*[A193V], respectively) and selected on LB agar plates containing 25 µg/ml lincomycin and 1 µg/ml erythromycin for macrolides (MLS) resistance. The obtained single recombinants were first verified using oligos specific for pMiniMad plasmid (oAR27 or oAR28) and flanking regions of *degU* (oAR25 and oAR26) or *degS* (oAR31 and oAR32) genes. The verified single recombinants were subsequently cultivated in liquid LB medium, cultures were plated on LB medium without antibiotics, single colonies were tested for the loss of integrated plasmid, and clones with specific SNPs (*degU*[S202G] and *degS*[A193V]) were selected. Plasmids pTB693 and pTB694 were obtained by cloning the PCR products obtained using oAR23 and oAR24 (*degU*[S202G] from genomic DNA extracted from CoEvo3) and oAR30 and oAR41 (*degS*[A193V] from genomic DNA extracted from CoEvo3) into plasmid pMiniMad[70] using restrictions enzymes NcoI and BamHI, and SalI and BamHI, respectively. Sequence of oligos are listed in Supplementary Table 1. Incorporation of the ancestor *degU* and *degS* alleles into the evolved variants was verified by sequencing. All other *B. subtilis* strains were constructed previously and the source is indicated in Supplementary Table 1.

The P_{agsA(3×RlmA-box)}-m*luc*-T*trpC* cassette with NotI sites was PCR-amplified using plasmid pBN008 as template and primers PagsAP1f-NotI and TtrpCP2r-NotI. Plasmid pBN008 contains the P_{agsA(3x rlmA-box)} fragment from plasmid P_{agsA}(0.55-kb-rlm2add)-*uidA-pyrG*[*52] and m*luc*-T*trpC* from plasmid pVG4.1[71]. The P_{agsA(3×RlmA-box)}-m*luc*-T*trpC* cassette with NotI sites was ligated into pJet1.2 and verified by sequencing. The

P_{agsA(3×RlmA-box)}-m*luc*-T*trpC* cassette was isolated with NotI and ligated into the NotI-digested plasmid pMA334[72], yielding pMA348. The mutation in the RlmA box of the *agsA* promoter to generate the P_{agsA(RlmA-box mutated)}-m*luc*-T*trpC* construct was introduced by PCR using primers PagsAP4f-NotI and PagsA-AF-R-mut-RlmA2 (211 bp) and primers PagsA-AF-F-mut-RlmA1 and PagsAP2r (441 bp) with genomic DNA from WT *A. niger* strain N402. Both PCR fragments were fused together by PCR with primers PagsAP4f-NotI and PagsAP2r (622 bp) and the fusion PCR product, containing a mutation in the RlmA box of the *agsA* promoter, was ligated into pJet1.2 and verified by sequencing. The mutated promoter was then isolated using NotI and EcoRI and ligated into corresponding sites of pBN008, yielding plasmid pMA368 P_{agsA(RlmA-box mutated)}-m*luc*-T*trpC*. The P_{agsA(RlmA-box mutated)}-m*luc*-T*trpC* cassette with NotI sites was PCR-amplified using plasmid pMA368 as template and primers PagsAP1f-NotI and TtrpCP2r-NotI, ligated into pJet1.2, and verified by sequencing. The P_{agsA(RlmA-box mutated)}-m*luc*-T*trpC* cassette was isolated using NotI and ligated into the NotI-digested plasmid pMA334[72], yielding pMA370. Plasmids pMA348 and pMA370 were digested with AscI to release the complete P_{agsA(3×RlmA-box)}-m*luc*-T*trpC-pyrG*** and P_{agsA(RlmA-box mutated)}-m*luc*-T*trpC-pyrG*** targeting cassettes and transformed into *A. niger* strain MA169.4[73], using fungal strains deficient in the non-homologous end-joining pathway[74]. Correct integration of the *pyrG*** targeting construct was confirmed by Southern blot for MA297.3 and MA584.2.

The *A. niger oahA* (NRRL3_06354) deletion strain was created using the split marker method with the *Aspergillus oryzae pyrG* gene (*AOpyrG*) as selection marker[75]. The flank regions of the *oahA* gene were PCR amplified with genomic DNA of N402 as template and primers oahAP1f and oahAP11r (5′ flank, 833 bp) or oahAP3f and oahAP4r (3′ flank, 1133 bp). The *pyrG* fragment was PCR amplified from plasmid pAO4-13[76] using primers AOpyrGP12f and AOpyrGP15r (5′ fragment, 1011 bp) or AOpyrGP14f and AOpyrGP13r (3′ fragment, 1248 bp). Split marker fragments were amplified by fusion PCR using oahA 5′ and AOpyrG 5′ as template and primers oahAP1f and AOpyrGP15r (5′ split marker fragment, 1826 bp) or oahA 3′ and AOpyrG 3′ as template and primers oahAP4r and AOpyrGP14f (3′ split marker fragment, 2376 bp) and column purified. The split marker fragments were transformed to *A. niger* strain MA169.4[73], according to ref. 74. Deletion of the *oahA* gene in strain MA824.1 was confirmed by PCR and lack of acidification on pH indicator plate.

Other *A. niger* strains were constructed previously and the source is indicated in Supplementary Table 1. Species identity of *Aspergillus* strains from the Jena Microbial Resource Collection (JMRC) was validated by sequencing their PCR-amplified calmodulin gene fragment[77] obtained using primers oBK7 and oBK8 (sequencing data included in Supplementary Dataset 3). Fusarium strains from the IBT Culture Collection were used as previously described[32].

## Bacterial evolution in the presence of the fungus

For co-cultivation, filtered conidia solution of *A. niger* was streaked on solid 1.5% agar containing LB medium in a hashtag pattern (Fig. 1a) using an inoculation loop. Diluted culture of *B. subtilis* (1:100) was streaked as a square through the inoculation lines of *A. niger* and the plate was incubated for 5 days at 28 °C. Small squares of the interaction zone of the bacterium and the fungus were excised and placed on a new LB plate to allow separation of *B. subtilis* by growing out from the old block of the medium over 1 day at 28 °C. Notably, no antifungal compound was added to the LB agar medium, the separation of the bacterium and the fungus was achieved simply due to the faster growth of the bacterium from the agar block. A sample of the isolated bacteria was collected and incubated overnight in liquid LB medium (shaking at 225 rpm), and the culture was used to incubate a new plate with freshly inoculated *A. niger*, and to create a cryo-stock that was stored at −80 °C. Throughout the experiment, dilution of the bacterial culture was increased stepwise to 1:1000 to prevent overgrowth of the

fungus by the evolved bacterial strains. As a control, *B. subtilis* was incubated without the fungus and the experiment was performed with five parallel replicates in each group. While the same procedure was applied for both treatments, we cannot exclude that the population size of the control linages was slightly higher due to absence of the fungus on the plates. The experiment was conducted for 10 weeks after which the evolved bacterial populations were clean streaked and two single isolates per replicate were kept for further analysis.

## Colony biofilm and surface spreading

To analyse changes in colony biofilm morphology, 2 μl of *B. subtilis* overnight culture was spotted on MSgg medium as previously described[78]. Images of colony biofilms were recorded using an Axio Zoom V16 stereomicroscope (Carl Zeiss, Jena, Germany). A Zeiss CL 9000 LED light source and an AxioCam MRm monochrome camera were employed (Carl Zeiss).

To examine the spreading behaviour of *B. subtilis* in the presence or absence of *A. niger*, two-compartment plates with LB medium and 1% agar were dried for 15 min and 2 μl of *B. subtilis* overnight culture was spotted in the middle of either one or both compartments. In the case of *A. niger* volatile challenge, fungal spores were inoculated in the other compartment 1 day before inoculation of bacterial cultures. The inoculated plates were dried for 10 min, incubated at 28 °C, and expansion of strains was measured after 1 day.

To test the effects of volatile compounds on colony spreading of *B. subtilis*, 5 μl of 500 μM solution of either pyrazine, dimethyl disulphide (DMS), 2-pentyene, or their mixture was spotted on 5 mm filter paper discs placed in one side of the two-compartment plates. Next, 5 μl of an overnight culture of ancestor or CoEvo3 isolate, adjusted to an optical density at 600 nm (OD600) of 1.0, was spotted in the second compartment and plates were dried for 10 min, sealed, and incubated for 96 h at 28 °C. Every 24 h, 5 μl of each compound or the mixture was added onto the paper disc. The *B. subtilis* colony size was recorded, and plates were photographed at the end of the experiment. The experiment was conducted twice with three biological replicates for each independent assay.

## Direct bacterium-fungus interaction on plates

Spreading of *B. subtilis* in the presence of the fungus was analysed by pre-culturing two streaks of the *Aspergillus* strain with a gap of 10 mm in between for 1 day at 28 °C. Afterwards, *B. subtilis* was spotted in the gap and the plates were incubated as described above.

To check for pH manipulation by *B. subtilis*, *A. niger* or other *Aspergillus* strains (see strain list and collection numbers in Supplementary Table 1), LB plates containing the pH indicator Bromocresol Purple (0.02 g/l) were prepared. The plates were inoculated with a streak of *Aspergillus* conidia in the middle next to 2 μl spots of *B. subtilis* culture and incubated at 28 °C for 2 days.

## Confocal laser scanning microscopy (CLSM)

Fungal cell morphologies were analysed using a confocal laser scanning microscope (LSM 780 equipped with an argon laser, Carl Zeiss, Jena, Germany) and Plan-Apochromat/1.4 Oil DIC (Differential Interference Contrast) M27 63 3 objective. Fluorescent reporter excitation was performed with the argon laser at 488 nm and the emitted fluorescence was recorded at 484–536 nm for GFP. Non-labelled *A. niger* hyphae didn't display detectable autofluorescens in our experiments with the microscopy settings used. Zen 2012 Software (Carl Zeiss) was used for both stereomicroscopy and CLSM (confocal laser scanning microscopy) image visualization.

## Population metagenome sequencing and variant calling

For whole-population sequencing, frozen stocks of the evolved bacterial populations and ancestor were cultured for 16 h at 37 °C and genomic DNA was extracted using a EURx Bacterial and Yeast Genomic DNA Kit. The culturing step was necessary to obtain sufficient material for metagenome sequencing from the frozen stock. Notably, the extra culturing might have introduced mutations at low frequencies into these samples. MGIEasy PCR-free Library Prep Set (MGI Tech) was used for creating acoustic fragmentation PCR-free libraries. Paired-end fragment reads (2 × 150 nucleotides) were generated on a DNBSEQ-Tx sequencer (MGI Tech) according to the manufacturer's procedures. Depth coverage >200× was obtained for all population samples before polymorphism calling.

Low-quality reads were first filtered from the raw data using SOAPnuke (version 1.5.6)[79], including reads with >50% of bases with a quality score <12, >10% of unknown bases (N), and adaptor contamination-containing reads. The similar variants calling sensitivity was ensured by normalisation of the clean data to 200× depth for all population samples. Mutations were then identified using breseq (version 0.35.7) with default parameters and the -p option for population samples[36,37]. The default parameters of breseq called mutations only if they appeared at least twice on each strand and reached a frequency of at least 5% in the population. The reference genome used for population resequencing analysis was the NCIB 3610 genome and the pBS plasmid (GenBank accession no. NZ_CP020102 and NZ_CP020103, respectively). Mutations that were also detected in ancestor strains and in high polymorphism regions were omitted from the list to create the final table of mutations.

## Genome resequencing of single evolved bacterial isolates

Genomic DNA of selected isolated strains was obtained as described above for evolved populations. An Illumina NextSeq sequencer was applied for generation of paired-end fragment reads (2 × 150 nucleotides) and bcl2fastq software (v2.17.1.14; Illumina) was applied for primary data analysis (base-calling). SOAPnuke (version 1.5.6)[79] was initially used for removing low-quality reads as described for the population samples. In addition, the first 10 bases of each read were removed. Mutations were called using breseq (version 0.35.7) with default parameters and the -p option[36,37]. Default parameters called mutations only if they appeared at least twice on each strand and reached a frequency of at least 5% in the sample. Mutation frequency data for each sequenced strain are included in Supplementary Dataset 1.

## Quantification of surfactin production

Bacterial colonies cultivated on LB medium with 1% agar for 2 days were harvested by collecting a plug of agar from the edge of a bacterial colony and subsequently sonicating and diluting for colony-forming counts. PTFE-filtered supernatant was used for semi-quantifying the amount of surfactin. An Agilent Infinity 1290 UHPLC system (Agilent Technologies, Santa Clara, CA, USA) equipped with a diode array detector was used for ultra-high-performance liquid chromatography-high-resolution mass spectrometry (UHPLC-HRMS). An Agilent Poroshell 120 phenyl-hexyl column (2.1 × 250 mm, 2.7 μm) was used for separation using a linear gradient consisting of water (A) and acetonitrile (B) both buffered with 20 mM formic acid, starting at 10% B and increasing to 100% over 15 min, holding for 2 min, returning to 10% B in 0.1 min, and holding for 3 min (0.35 ml/min, 60 °C). A 1 μl injection volume was applied. Mass spectra were recorded on an Agilent 6545 QTOF instrument equipped with an Agilent Dual Jet Stream electrospray ion source in positive ion mode as centroid data in the range *m/z* 85–1700, with an acquisition rate of 10 spectra/s. A drying gas temperature of 250 °C, gas flow of 8 l/min, sheath gas temperature of 300 °C and flow of 12 l/min were used. The capillary voltage and nozzle voltage were set to 4000 V and 500 V, respectively. Surfactin A and its isoforms ($C_{53}H_{93}N_7O_{13}$) were semi-quantified using MassHunter quantitative analysis B.07.00 with the $[M+H]^+$ ion (*m/z* 1036.6904) and an isolation window of 10 ppm. A calibration curve was constructed using an authentic standard (Sigma-Aldrich).

Semi-quantitative comparison of surfactin production in the ancestor, CoEvo2, and *degU*[S202G] strains has been performed using drop collapse assays as previously[68]. Briefly, 5 µl supernatant of cell-free overnight bacterial culture was spotted onto parafilm next to a ruler used for scale, the droplets were photographed after 5 min using an iPhone 13 mini, and the size of the droplet was measured using "Measure" function of Image J (v1.54).

## MALDI-MSI

Petri dishes (5 cm diameter) were filled with ~4.5 mL of 1% LB and allowed to dry for 10 min. Subsequently, spores of *A. niger* were inoculated with a plastic loop drawing two separated lines and leaving a non-inoculated area at the centre of the plate. After 24 h of incubation at 28 °C, 5 µL of *B. subtilis* ancestor, CoEvo2 or Δ*srfAC* were spotted onto the non-inoculated area. The plates were sealed with parafilm and incubated for 96 h. Samples were excised and mounted on an IntelliSlides conductive tin oxide glass slide (Bruker Daltonik GmbH) precoated with 0.25 ml of 2,5-dihydrobenzoic acid (DHB) by an HTX Imaging TM-Sprayer (HTX Technologies, USA). Slide images were subsequently taken using TissueScout (Bruker Daltonik GmbH) and a Braun FS120 scanner, followed by overnight drying in a desiccator. Subsequently, samples were overlayed by spraying 1.75 ml of DHB (20 mg/ml in ACN/MeOH/H2O (70:25:5, v/v/v)) in a nitrogen atmosphere and dried overnight in a desiccator prior to MSI measurement. Samples were analysed using a timsTOF flex mass spectrometer (Bruker Daltonik GmbH) for MALDI MSI acquisition in positive MS scan mode with 100 µm raster width and a mass range of 100 – 2000 Da. Calibration was performed using red phosphorus. The following settings were used in timsControl. Laser imaging 100 µm, Power Boost 3.0%, scan range 26 µm in the XY interval, laser power 90%; Tune Funnel 1 RF 300 Vpp, Funnel 2 RF 300 Vpp, Multipole RF 300 Vpp, isCID 0 eV, Deflection Delta 70 V, MALDI plate offset 100 V, quadrupole ion energy 5 eV, quadrupole loss mass 100 m/z, collision energy 10 eV, focus pre-TOF transfer time 75 µs, pre-pulse storage 8 µs. After data acquisition, SCiLS software was used for data analysis and all data were root mean square normalised.

## Volatile assay and trapping of VOCs

For analysis and trapping of VOCs, two-compartment glass Petri-dishes[80] containing 1% LB agar were employed. *A. niger* was inoculated 24 h before addition of the bacterial strain on the left side of the two-compartment glass Petri dish and incubated overnight at 28 °C (Supplementary Fig. 2). After CoEvo 3 was incubated overnight at 28 °C in liquid LB medium, 2 µl of culture was spotted on the opposite compartment of the fungal culture. The glass Petri dishes were kept open for 25 min to allow droplets to dry. Plates were then closed by a lid with an outlet connected to a steel trap containing 150 mg Tenax TA and 150 mg Carbopack B (Markes International Ltd., Llantrisant, UK) and incubated at 28 °C. The Tenax steel traps were collected after 3 and 7 days of incubation and stored at 4 °C until GC-Q-TOF analysis. Glass Petri dishes containing LB agar medium but without inoculated bacteria or fungi served as controls.

## GC-Q-TOF measurement and volatile analysis

For trapping and analysing VOCs[81], volatiles were desorbed from traps using a Unity TD-100 desorption unit (Markes International Ltd.) at 210 °C for 12 min (He flow 50 ml/min) and trapped on a cold trap at −10 °C. Volatiles were introduced into the GC-Q-TOF (Agilent 7890B GC and Agilent 7200 A QTOF; Agilent Technologies) by heating the cold trap for 12 min to 250 °C. The split ratio was 1:10 and the column was a 30 × 0.25 mm ID RXI-5MS with a film thickness of 0.25 µm (Restek 13424-6850, Bellefonte, PA, USA). The temperature programme was as follows: 39 °C for 2 min, 39 °C to 95 °C at 3.5 °C/min, 95 °C to 165 °C at 4 °C/min, 165 °C to 280 °C at 15 °C/min, 280 °C to 320 °C at 30 °C/min, holding for 7 min. VOCs were ionised in EI mode at eV and mass spectra

were acquired in full scan mode (30 – 400 U @ 5 scans/s). Mass spectra were extracted with MassHunter Qualitative Analysis Software V B.06.00 Build 6.0.633.0 (Agilent Technologies). The obtained mass spectra were transformed to netCDF files and imported into MZmine V2.20 (Copyright 2005−2012) MZmine Development Team)[82]. Compounds were identified via their mass spectra using the deconvolution function and local minimum search algorithm in combination with two mass spectral libraries, NIST 2014 V2.20 (National Institute of Standards and Technology, USA, http://www.nist.gov) and Wiley 7th edition spectral libraries, and by their linear retention index (LRI) calculated using AMDIS 2.72 (National Institute of Standards and Technology, USA). After deconvolution and mass identification, peak lists containing the mass features of each treatment were exported as csv files for further analysis.

## Statistical analyses

Statistical analysis of volatile metabolite data was performed using MetaboAnalyst V3.0 (www.metaboanalyst.ca)[83]. Prior to statistical analysis, data normalisation was performed via log-transformation and data were mean-centred and divided by the standard deviation of each variable. To identify significant mass features, one-way analysis of variance (ANOVA) with post-hoc TUKEY test (HSD-test) and PLSD analysis were performed between datasets. Mass features were considered statistically significant at $p \leq 0.05$.

One-way ANOVA was conducted to compare the effect of pure compounds and the mixture on the colony size of CoEvo3. Tukey's HSD test was used to determine significant differences between means, which are denoted by letters. For comparing the acidified area and surfactin in drop collapse assay significant variations in means of different groups to the ancestor were assessed using analysis of variance (ANOVA), along with Tukey−Kramer's post hoc analysis.

To evaluate differences in surfactin production between strains and to explore the influence of *rlmA* mutation in *A. niger*, Student's *t*-test with Bonferroni−Holm correction was performed. Student's unpaired two-tailed *t*-test was used for the Jaccard index.

## Reporting summary

Further information on research design is available in the Nature Portfolio Reporting Summary linked to this article.

# Data availability

Population sequencing data have been deposited in NCBI Sequence Read Archive (SRA) database under BioProject accession numbers PRJNA1107377; and in the CNGB Sequence Archive (CNSA)[84] of the China National GeneBank DataBase (CNGBdb)[85] under accession number CNP0003923 and for the DK1042 ancestor strain under CNP0002416. Isolate sequencing data have been deposited at the NCBI Sequence Read Archive (SRA) database under BioProject accession numbers PRJNA625867 (ancestor) and PRJNA926387 (evolved clones). VOC data available on Metabolomics Workbench as project PR001621 (https://doi.org/10.21228/M85M6X). Data points are all included in Supplementary Datasets. All other data generated and analysed during this study are either available in Source Data or can be requested from the corresponding author. Source data are provided with this paper.

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

## Acknowledgements

This work was supported by the Danish National Research Foundation (DNRF137) for the Centre for Microbial Secondary Metabolites to Á.T.K. and T.O.L., the Novo Nordisk Foundation within the INTERACT project of the Collaborative Crop Resiliency Programme (NNF19SA0059360) to

Á.T.K., and a start-up fund from Institute of Biology Leiden to Á.T.K. Funding was provided by the Novo Nordisk Foundation for infrastructure "Imaging Microbial Language in Biocontrol (IMLiB)" (NNF19OC0055625) to Á.T.K. and T.O.L.; G.H. was supported by the China National GeneBank (CNGB). J.X. was supported by a China Scholarship Council fellowship. The authors thank Aaron J.C. Andersen and the DTU Bioengineering Metabolomics Core for support with LC-MS and MALDI-MSI.

## Author contributions

Á.T.K. conceived the project. A.R., F.B., J.W.S., B.K., S.S., J.X., and Á.T.K. performed the experiments. G.H. and Y.W. performed bacterial population metagenomics experiments and corresponding data analysis. G.M. performed genome resequencing of the evolved bacterial clones. S.A.J. and M.W. performed MALDI-MSI. O.T. and P.G. performed VOC identification. C.N.L.A. performed microbiology tests on VOCs. X.X. performed analysis and statistics. C.B.W.P. performed LC-MS on lipopeptides. T.O.L. contributed methods and instrumentation for lipopeptide analysis. M.A., A.F.J.R and C.A.M.v.d.H. created fungal strains. T.J. and Á.T.K. wrote the manuscript with corrections from all authors.

## Competing interests

The authors declare no competing interests.
