## [Peer Review File · Nature Communications]

Enhanced surface colonisation and competition during bacterial adaptation to a fungusReviewer #1 (Remarks to the Author):

This manuscript uses a simple, yet elegant co-culture system to explore how interactions between a model bacterium (*B. subtilis*) and fungus (*A. niger*) affect their evolution. There has been an explosion of research on bacterial-fungal interactions (BFIs) in the past decade, but most of this work focuses on short-term ecological outcomes. The longer-term evolutionary consequences of BFIs have not been extensively explored. There are quite a few other recent studies demonstrating that the ecological context for a microbial species can impact how that microbe evolves. So demonstrating that species interactions affect microbial evolution is not novel. But this work is some of the first to demonstrate at a mechanistic level how BFIs affect microbial evolution. The main finding is that *B. subtilis* can evolve when growing with *A. niger* to have traits that negatively affect the *A. niger*. The authors do a fantastic job diving into the mechanisms of adaptation using genetic tools, microscopy, and other assays. They do this not only for the bacterium, but also for the fungus (the bulging phenotype). I am impressed with the large number of mechanistic datasets that were packed into this rich manuscript.

At the same time, I found that the manuscript was a bit bloated and had some datasets that didn't really seem to fit together naturally. For example, the volatile data are fascinating, but I am not sure why the authors decided to focus on volatiles and the work seems tangential. It is also incomplete because these data were not collected for the ancestor strain. I would remove this section from the manuscript because it does not strengthen the manuscript and there is plenty of other important mechanistic data that makes more sense.

The surfactin work is interesting and there is some fantastic genetic and microscopy work showing how *B. subtilis* surfactants can induce an interesting bulging phenotype in the *A. niger*. But it was unclear to me how any of the surfactin work related to the evolution experiment. Were there specific mutations in surfactin production pathways that could contribute to differences in surfactin production observed in the CoEvo2 and CoEvo3 strains? Is *degU* known to regulate surfactin production? I don't see any *srfAC* or *sfp* mutations in the evolution experiment, so I am confused how there is a bridge between the first part of the paper and the surfactin section. This was all unclear to me and as with the volatiles, the surfactin story felt a bit like a tangent.

I suggest that the authors consider breaking this manuscript into smaller pieces and submit them as individual papers (1. Evolution story, 2. Volatile story... would need more experiments/data, and 3. Surfactin story).

Another major limitation of this work is a lack of a clear ecological context where these organisms would interact and a lack of a connection to real world genotypes and phenotypes. The findings are really restricted to laboratory conditions and this model interaction. This BFI may occur in soils (the authors do note that *B. subtilis* and *A. niger* do co-occur in nature in soil), but there would also be many other species interacting with this pair, potentially diffusing and direct pairwise interactions between the two species. I appreciate the simplified model system on lab media and the use of model organisms to be able to get at genetic mechanisms, but to have greater impacts and relevance beyond the Petri dish, I would like to see how this work connects to natural populations of the interacting microbes.

The introduction wanders a lot and contains several sections of text that are not relevant to the background. For example, there is a whole paragraph about how bacteria can spread on mycelial networks, but this is not related to the current work. The paragraph starting in line 70 that explores how bacteria can upregulate secondary metabolite production in fungi also seems out of place and not relevant to the findings. The intro also lacks any type of predictions or framework for making predictions. What genes or traits did the authors think might have evolved in *B. subtilis* when it was co-cultured with the fungus? How might the fungus provide a selective environment that could shape the evolution of the bacterium?

The idea of "increased niche colonization" is used in the title and a few places in the manuscript, but it is not clear what the authors mean by this. In ecology, increased niche colonization would probably translate to the evolved microbe having the ability to access a broad set of environments or resources. But really what the authors observed was an increased spreading phenotype. This

would be increased dispersal/migration or simply the ability to use more space and outcompete your neighbor, and not increased niche colonization. This needs to be reframed or better explained.

I am a bit confused by some aspects of the experimental design of the co-culture evolution experiment. The authors note that they started to dilute the bacterial inoculum as the fungus started to become outcompeted by the bacterial mutants. How did this affect bacterial population sizes? Was the same dilution approach used for both the conditions "B. subtilis alone" and "B. subtilis + A. niger" populations? Can population sizes be carefully controlled or quantified using the hashtag streak setup and the agar plug transfer approach?

Related to population sizes, how many generations of B. subtilis were there during the 10 week evolution experiment?

Line 106: Was there antifungal added to the medium to exclude fungal growth? Could this step of removing the fungus add an artificial selection step for growth in liquid media?

Line 463: Why were only two isolates per replicate selected? What is the rationale for not selecting more? Did this capture the full range of colony morphologies that were observed?

Line 525: I am confused by the phrasing: "and reached a frequency of at least 5% in the sample, and mutations with a frequency <50% were removed." That seems to be saying that you kept mutations that were >5% but then tossed those that were <50%? Perhaps you are describing two different types of frequencies and the wording is just confusing. This needs to be clarified.

Figure 1C: How were these colonies selected? Does this span the full scope of morphotypes and mutations that were observed in the work? (see note above)

Line 499: Does the step where you grow up the populations from the frozen stocks change the frequencies of mutants in the populations? Why not extract DNA directly from these frozen stocks?

The paragraph starting on line 150 is really long and should be broken into two.

Reviewer #2 (Remarks to the Author):

The manuscript titled "Enhanced niche colonisation and competition during bacterial adaptation to a fungus" by Anne Richter et al. investigate the bacterial-fungal interactions in the context of bacterial evolution in the long-term co-cultivation experiment. Furthermore, authors hypothesize that biosurfactants production by the bacteria could be potentially disturb fungal expansion and acidification on the medium.

Overall, the idea of the project is extremely interesting as it addresses the crucial role of bacterial adaption to the fungus in an experimental evolution method. This could potentially help to understand how soil bacterial-fungal interactions evolve over the time.

Nonetheless several critical points were missing for the manuscript to reach appropriate conclusions. The main point of criticism, in my view, concerns the representativeness of the results obtained. If I understand well the design, 5 replicates per treatment (bacteria alone vs bacteria/fungi) were done and out of those 2 gave a significant phenotype for the BFI treatment but none for the bacteria alone treatment. Knowing the probability of a mutation occurring, is such a small number of replicas enough to draw a clear conclusion that the phenomenon is not random but due to the interaction with the fungus? What's more, the phenotypic analyses are based on images of plates that have never appeared to have been made in exactly the same way (different numbers of bacterial spots from one plate to another, unequal streak sizes, presence of aerosol fungal contaminants), which gives an impression of poor quality work, in addition to the fact that

there is no quantification and statistical analysis of the phenotypes, although this should be possible today.

In addition, the flow of the manuscript would need improvement to ease its reading and the interpretation of data. The introduction provides a clear description of the knowledge background about fungal-bacterial interactions but nothing regarding bacterial evolution while this question is at the heart of the study. The hypotheses that the authors want to test are not clearly stated, and we move from one section of results to another without clear links or a clear explanation of the scientific logic behind the experiments, which gives the impression of pieces of studies that have been aggregated rather than a set of experiments carried out to answer a clear question. The manuscript would clearly gain in clarity if a common thread was stated and followed. Last, critical details on the experimental set ups and methodology used are often missing, which makes it difficult to clearly understand what has been done and/or how it can be interpreted.

Reviewer #3 (Remarks to the Author):

In this manuscript Richter and collaborators explore the way a *Bacillus* strain evolves during the interaction with the fungus *Aspergillus niger*. I do feel the manuscript is succinct, well written, and balanced in terms of main figures and supplemental information. It is noteworthy that they find mutations in the two component systems DegS-DegU that seems to be related to the phenotype of the author called evolved strains. Efforts are also valuable when trying to explain the mechanistic behind the main finding that give title to this piece of work, spreading, which appears to be related to the production of surfactin.

There are some comments I would like to share with the authors.

1.- It wasn't clear to me why the interaction *Bacillus-Aspergillus*, and I assumed that both strains were isolated from the same samples. Then, in the introduction it is said that *Bacillus* and *Aspergillus* are found in soil, and thus, they might potentially co-habit. Even that interesting in terms of mechanistic ecological studies or evolution, it would be more convenient if this study had been conducted with some co-isolated strains, then it would be more feasible to transfer the findings of the study to a real scenario. In this sense, something have been said on *Fusarium* in this study, which even that not cohabitant with *Bacillus*, or maybe, at least they can "meet" in soils in agricultural systems.

2.- *Aspergillus* is acidifying the medium, and this is something common to other if not most of fungi. How relevant acidification is the evolution of the interaction? Is it possible that acidifying the medium *Bacillus* may evolve accumulating mutations in DegS-DegU?

In the other hand, what appears to happen in this interaction is that antagonism transit to a more antagonistic interaction, even that not much is said from the fungal side.

3. Related to the acidification and the buffering. Looking the images, it is not clear to me that CoEvo2 is the only strain that restrict acidification. Indeed, Bac1, 4 or 5 also phenocopy this action. I would like knowing what is this finding really adding to the rationale of the study.

4. When talking on the spreading of the evolved strains, which is the medium used in the experiments of interaction *Bacillus-Aspergillus*? It is said that CoEvo3 spread more in the presence of the fungus, but I missed how the colony looks in the same medium alone.

5. It is really interesting the effect of the volatiles on the spreading of the colonies. However, I do feel that it is kind of superficial at this point, and that could be removed from the study without affecting to the main message. With all the information at genomic level, and the battery of mutants collected it could be explore which pathways are affected and whether they are somehow related to DegS-DegU.

6. It appears that the spreading phenotype of on the the coevolved strains is related to the overproduction of surfactin. Did the auhtor have found some realtion of surfactin overproduction and the mutation in the DEG system? or other wise these are complementary phenotypes. It would be compelling trying to find a putative connection between them. Given that clean mutants have been made on this loci, I consider relevant to analyze the level of production of surfactin and at least plipastatin in this builded strains, a finding that would help reinforcing the conclusion. Have the authors investigated how are the biofilm related genes expressed in the evolved strain?

Although it appears that the role of surfactin in triggering biofilm formation is strain dependent, adding some information would help understanding the spreading phenotype (directly related to surfactin, or indirectly).

7. In the study of the damage inflicted to *Aspergillus*, it is concluded that surfactin is the main responsible for the antagonistic interaction. First, it surprised me the absence of effect of plipastatin, however I do not really know if plipastatin is not antifungal or simply it is failing in the induction of budding. Second, in the luciferase experiments, I missed the effect of pure surfactin or plipastatin to support the findings at macroscopic or microscopic level. The experiments where surfactin is shown to modify the fungal hyphae has been done in liquid or solid? What happened if surfactin is added in solid medium inoculated with *Aspergillus*? In this experiment we might find a similar effect with no interference related to bacterial cells.

Third, how conserved is the pathway activated in *Aspergillus* in other fungi? If pretty well conserved, then one would expect a similar response in *Fusarium*, which is not.

8. Surfactin and other CLPs are known to target membranes but not the cell wall. Also, the authors have shown some experiments suggesting alterations of membranous systems but not cell wall. Thus I would not conclude on a direct action surfactin on the cell wall. That effect can be the conclusion of a sequence of events affecting the cell membrane and other cytological disorders. I would like to see some ultrastructural studies of fungal cell, to really appreciate the alterations suggested with the fluorescent probes.

Is it surfactin killing *Aspergillus* cells, or it is only inducing these morphological changes? I think I did not see any experiments on viability or antifungal activity. Surfactin is not among the most potent antifungal CLPs produced by *Bacillus*.

Reviewer #1 (Remarks to the Author):

This manuscript uses a simple, yet elegant co-culture system to explore how interactions between a model bacterium (*B. subtilis*) and fungus (*A. niger*) affect their evolution. There has been an explosion of research on bacterial-fungal interactions (BFIs) in the past decade, but most of this work focuses on short-term ecological outcomes. The longer-term evolutionary consequences of BFIs have not been extensively explored. There are quite a few other recent studies demonstrating that the ecological context for a microbial species can impact how that microbe evolves. So demonstrating that species interactions affect microbial evolution is not novel. But this work is some of the first to demonstrate at a mechanistic level how BFIs affect microbial evolution. The main finding is that *B. subtilis* can evolve when growing with *A. niger* to have traits that negatively affect the *A. niger*. The authors do a fantastic job diving into the mechanisms of adaptation using genetic tools, microscopy, and other assays. They do this not only for the bacterium, but also for the fungus (the bulging phenotype). I am impressed with the large number of mechanistic datasets that were packed into this rich manuscript.

At the same time, I found that the manuscript was a bit bloated and had some datasets that didn't really seem to fit together naturally. For example, the volatile data are fascinating, but I am not sure why the authors decided to focus on volatiles and the work seems tangential. It is also incomplete because these data were not collected for the ancestor strain. I would remove this section from the manuscript because it does not strengthen the manuscript and there is plenty of other important mechanistic data that makes more sense.

The surfactin work is interesting and there is some fantastic genetic and microscopy work showing how *B. subtilis* surfactants can induce an interesting bulging phenotype in the *A. niger*. But it was unclear to me how any of the surfactin work related to the evolution experiment. Were there specific mutations in surfactin production pathways that could contribute to differences in surfactin production observed in the CoEvo2 and CoEvo3 strains? Is *degU* known to regulate surfactin production? I don't see any *srfAC* or *sfp* mutations in the evolution experiment, so I am confused how there is a bridge between the first part of the paper and the surfactin section. This was all unclear to me and as with the volatiles, the surfactin story felt a bit like a tangent.

The surfactin section explain the influence of surfactin on *A. niger*. We now provide experimental evidence that an evolved isolate and the *degU*^{S202G} mutant delay the fungal growth, which might be explained by the detected enhanced surfactin production. We demonstrate that surfactin production is enhanced in CoEvo2 (Fig 3a), the *srfAC* mutant allows higher degree of acidification by the fungus (Supplementary figure 2a and b), and CoEvo2 and *degU*^{S202G} strains reduce fungal growth compared to the ancestor, therefore we set out to examine the influence of surfactin on *A. niger*. These are introduced in the results chapter describing the influence of surfactin on the fungus.

I suggest that the authors consider breaking this manuscript into smaller pieces and submit them as individual papers (1. Evolution story, 2. Volatile story... would need more experiments/data, and 3. Surfactin story).

To simplify the story line, the volatile experiments have been moved fully to the supplementary material due to its preliminary aspects. However, these preliminary results explain why CoEvo3 and *degS*^{A193V} mutant does not spread in the presence of the fungus, but it has increased surface colonization in the absence of the fungus; thus, these explanations were retained in the supplementary material.

Another major limitation of this work is a lack of a clear ecological context where these organisms would interact and a lack of a connection to real world genotypes and phenotypes. The findings are really restricted to laboratory conditions and this model interaction. This BFI may occur in soils (the authors do note that *B. subtilis* and *A. niger* do co-occur in nature in soil), but there would also be many other species interacting with this pair, potentially diffusing and direct pairwise interactions between the two species. I appreciate the simplified model system on lab media and the use of model

organisms to be able to get at genetic mechanisms, but to have greater impacts and relevance beyond the Petri dish, I would like to see how this work connects to natural populations of the interacting microbes.

We agree with the reviewer, the direct ecological context is missing here as it was not an aim of our study. However, we provide a conceptual laboratory experimental evolution approach that can be utilized to improve antifungal properties of bacteria, as we expected bacterial clones being selected that have increased competition against the fungus due to potential resource competition.

The introduction wanders a lot and contains several sections of text that are not relevant to the background. For example, there is a whole paragraph about how bacteria can spread on mycelial networks, but this is not related to the current work. The paragraph starting in line 70 that explores how bacteria can upregulate secondary metabolite production in fungi also seems out of place and not relevant to the findings. The intro also lacks any type of predictions or framework for making predictions. What genes or traits did the authors think might have evolved in *B. subtilis* when it was co-cultured with the fungus? How might the fungus provide a selective environment that could shape the evolution of the bacterium?

The introduction explains the different type of bacterial-fungal interactions: how secondary metabolites can be enhanced in the presence of the other partner, which we indeed observe in adapted *B. subtilis* isolates, how their interaction contributes to motility (i.e. spreading of the bacteria), which again seems to be influence in certain evolved isolates. We have now included hypothesized predictions based on these previous works.

The idea of “increased niche colonization” is used in the title and a few places in the manuscript, but it is not clear what the authors mean by this. In ecology, increased niche colonization would probably translate to the evolved microbe having the ability to access a broad set of environments or resources. But really what the authors observed was an increased spreading phenotype. This would be increased dispersal/migration or simply the ability to use more space and outcompete your neighbor, and not increased niche colonization. This needs to be reframed or better explained.

We agree with the reviewer. We now indicate enhanced surface colonization in accordance with the increased spreading of the specific isolates.

I am a bit confused by some aspects of the experimental design of the co-culture evolution experiment. The authors note that they started to dilute the bacterial inoculum as the fungus started to become outcompeted by the bacterial mutants. How did this affect bacterial population sizes? Was the same dilution approach used for both the conditions “*B. subtilis* alone” and “*B. subtilis* + *A. niger*” populations? Can population sizes be carefully controlled or quantified using the hashtag streak setup and the agar plug transfer approach?

The same dilution and inoculation procedure were used for both conditions Bac and CoEvo lines. We cannot exclude that Bac lineages contained slightly higher population size, as the fungus was absent under those conditions, therefore there was no competition for the resources. This is now mentioned in the materials and methods. We do not have the exact population size determined from these experiments, thus we cannot report precise generation numbers, as in our previous publications with biotic interactions (Blake et al 2021 *Environ Microbiol* doi: 10.1111/1462-2920.15680; Lin et al 2021 *mSystems* doi: 10.1128/mSystems.00864-21; Nordgaard et al 2022 *iScience* doi: 10.1016/j.isci.2022.104406; Hu et al 2023 *Microbial Genomics* doi: 10.1099/mgen.0.001064; and Hu et al 2023 *mSystems* doi: 10.1128/msystems.00548-23).

Related to population sizes, how many generations of *B. subtilis* were there during the 10 week evolution experiment?

Unfortunately, we cannot estimate the generation number. We cannot estimate the number of generations of the plates, where we use a streak-out of the diluted culture and let them grow in the presence of the fungus, where only a small area was extracted. In the lack of the exact bacterial CFUs is each step, any estimation would be far away from the actual generation number and population sizes. This is a parameter that is not available in the absence of exact

population size being established in the experimental evolution systems, see for example in our previous publications: Blake et al 2021 *Environ Microbiol* doi: 10.1111/1462-2920.15680; Lin et al 2021 *mSystems* doi: 10.1128/mSystems.00864-21; Nordgaard et al 2022 *iScience* doi: 10.1016/j.isci.2022.104406; Hu et al 2023 *Microbial Genomics* doi: 10.1099/mgen.0.001064; and Hu et al 2023 *mSystems* doi: 10.1128/msystems.00548-23.

Line 106: Was there antifungal added to the medium to exclude fungal growth? Could this step of removing the fungus add an artificial selection step for growth in liquid media?

No antifungal compound was added. However, due to the faster growth, the bacterium was able to expand faster on the new LB agar medium. This information is now indicated in the material and methods. The liquid growth was required to obtain enough cells for the next round of the inoculation. We have first tested approaches that allow transfer of both the bacterium and the fungus to the next generation, however, all our efforts failed. The only possibility we were successful to obtain the co-cultures on the plates was to inoculate the diluted overnight bacterial culture next to the fungus that was initiated using the fungal spores.

Line 463: Why were only two isolates per replicate selected? What is the rationale for not selecting more? Did this capture the full range of colony morphologies that were observed?

We did not observe broad morphology diversification in these lineages as we observed previously (e.g. Blake et al 2021 *Environ Microbiol* doi: 10.1111/1462-2920.15680 or Hu et al 2023 *Microbial Genomics* doi: 10.1099/mgen.0.001064), therefore only two replicates were selected randomly. The replicates phenocopy each other when colony morphology is examined except one pair in one of the control samples (Bac 3.1 and Bac 3.2).

Line 525: I am confused by the phrasing: "and reached a frequency of at least 5% in the sample, and mutations with a frequency <50% were removed." That seems to be saying that you kept mutations that were >5% but then tossed those that were <50%? Perhaps you are describing two different types of frequencies and the wording is just confusing. This needs to be clarified.

Thank you for noticing this. Indeed, we used the at least 5% cut off. The sentence has been corrected.

Figure 1C: How were these colonies selected? Does this span the full scope of morphotypes and mutations that were observed in the work? (see note above)

The full set of biofilm colonies are included the supplementary figures, few representatives are displayed in Fig 1C to demonstrate the remarkable difference between CoEvo and Bac evolved isolates.

Line 499: Does the step where you grow up the populations from the frozen stocks change the frequencies of mutants in the populations? Why not extract DNA directly from these frozen stocks?

We have limited volume of the frozen stock remaining from the populations, thus for sufficient DNA preparation, we had to perform a cultivation step. We agree that this might introduce additional mutation and this possibility is now indicated in the materials and methods.

The paragraph starting on line 150 is really long and should be broken into two.

The indicated paragraph has been broken into three sub sections: populations metagenome sequencing, isolate genome sequencing, and reintroduction of the mutations. However, we prefer to keep one heading for these, as these paragraphs all refer to the mutations described and validated.

Reviewer #2 (Remarks to the Author):

The manuscript titled "Enhanced niche colonisation and competition during bacterial adaptation to a fungus" by Anne Richter et al. investigate the bacterial-fungal interactions in the context of bacterial evolution in the long-term co-cultivation experiment. Furthermore, authors hypothesize that biosurfactants production by the bacteria could be potentially disturb fungal expansion and

acidification on the medium.

Overall, the idea of the project is extremely interesting as it addresses the crucial role of bacterial adaptation to the fungus in an experimental evolution method. This could potentially help to understand how soil bacterial-fungal interactions evolve over the time.

Nonetheless several critical points were missing for the manuscript to reach appropriate conclusions. The main point of criticism, in my view, concerns the representativeness of the results obtained. If I understand well the design, 5 replicates per treatment (bacteria alone vs bacteria/fungi) were done and out of those 2 gave a significant phenotype for the BFI treatment but none for the bacteria alone treatment. Knowing the probability of a mutation occurring, is such a small number of replicas enough to draw a clear conclusion that the phenomenon is not random but due to the interaction with the fungus? What's more, the phenotypic analyses are based on images of plates that have never appeared to have been made in exactly the same way (different numbers of bacterial spots from one plate to another, unequal streak sizes, presence of aerosol fungal contaminants), which gives an impression of poor quality work, in addition to the fact that there is no quantification and statistical analysis of the phenotypes, although this should be possible today.

We have now reworded the results and the conclusions as indicated by the reviewer not to claim that the evolutionary path is only possible in the presence of the fungus, but that such differentiation has been observed in the presence of the fungus in one population. We thank the Reviewer for this comment to be more precise with our interpretation. We have performed all phenotyping experiments with at least 3 replicates, this is now indicated in the materials and methods. The bacterial spots were the same within each experiment, however, due to the overgrowth of the fungus, some spots are not visible. However, the pH indicator plate and colony expansion experiments required different experimental setup for visible changes; these were adapted accordingly. We have now added quantification of the low pH area and made quantitative estimation of anti-*Aspergillus* activity of the ancestor, CoEvo2, and *degU*^{S202G} strains. Unfortunately, the architecturally complex colony biofilm is a qualitative measure of biofilm formation, this is a generally used approach in the *Bacillus* field.

In addition, the flow of the manuscript would need improvement to ease its reading and the interpretation of data. The introduction provides a clear description of the knowledge background about fungal-bacterial interactions but nothing regarding bacterial evolution while this question is at the heart of the study. The hypotheses that the authors want to test are not clearly stated, and we move from one section of results to another without clear links or a clear explanation of the scientific logic behind the experiments, which gives the impression of pieces of studies that have been aggregated rather than a set of experiments carried out to answer a clear question. The manuscript would clearly gain in clarity if a common thread was stated and followed. Last, critical details on the experimental set ups and methodology used are often missing, which makes it difficult to clearly understand what has been done and/or how it can be interpreted.

We apologize for the brevity of our explanations; this was caused by the brief format we adhered when preparing the manuscript to another Nature journals. We have now extended the explanations based on the Reviewers' specific suggestions. As Reviewer 1 suggested, we added predictions what we expect during bacterial adaptation. We also tried to connect the different experiments. Experimental methods have been added where commented by the reviewers.

Reviewer #3 (Remarks to the Author):

In this manuscript Richter and collaborators explore the way a *Bacillus* strain evolves during the interaction with the fungus *Aspergillus niger*. I do feel the manuscript is succinct, well written, and balanced in terms of main figures and supplemental information. It is noteworthy that they find mutations in the two component systems DegS-DegU that seems to be related to the phenotype of the author called evolved strains. Efforts are also valuable when trying to explain the mechanistic behind the main finding that give title to this piece of work, spreading, which appears to be related to the production of surfactin.

There are some comments I would like to share with the authors.

1.- It wasn't clear to me why the interaction Bacillus-Aspergillus, and I assumed that both strains were isolated from the same samples. Then, in the introduction it is said that Bacillus and Aspergillus are found in soil, and thus, they might potentially co-habit. Even that interesting in terms of mechanistic ecological studies or evolution, it would be more convenient if this study had been conducted with some co-isolated strains, then it would be more feasible to transfer the findings of the study to a real scenario. In this sense, something have been said on Fusarium in this study, which even that not cohabitant with Bacillus, or maybe, at least they can "meet" in soils in agricultural systems.

These strains were not isolated from the same samples, both *B. subtilis* 3610 and *A. niger* N402 were isolated several decades ago and used as model organisms for the respective species. *B. subtilis* group species are being used commercially as biocontrol agent against *Fusarium*, hence we tested if *B. subtilis* similarly influence the mycelia morphology. This insight will be relevant for those working on biocontrol.

2.- Aspergillus is acidifying the medium, and this is something common to other if not most of fungi. How relevant acidification is the evolution of the interaction? Is it possible that acidifying the medium Bacillus may evolve accumulating mutations in DegS-DegU?

This is a very interesting suggestion, we are not aware of any study that would have evolved *B. subtilis* in the presence of weak acids, therefore, we cannot answer this question. Here, using the *A. niger oahA::AOpyrG* strain that is unable to acidify the medium, we demonstrate that in the lack of acidification, *B. subtilis* ancestor, as well as CoEvo2, *degU*^{S202G}, and *srfAC* strains colony growth is comparable to the conditions when acidification is observed (Supplementary figure 2c). This, suggest that acidification does not influence the colony growth and morphology of *B. subtilis*. This is now indicated in the manuscript.

In the other hand, what appears to happen in this interaction is that antagonism transit to a more antagonistic interaction, even that not much is said from the fungal side.

We agree with the reviewer on this interpretation.

3. Related to the acidification and the buffering. Looking the images, it is not clear to me that CoEvo2 is the only strain that restrict acidification. Indeed, Bac1, 4 or 5 also phenocopy this action. I would like knowing what is this finding really adding to the rationale of the study.

We have now quantified the degree of acidification by measuring the yellow area of the plate. We exchanged the images for representative samples that are easier to observe the differences after 2 days of incubation. The quantification of yellow colored area now allowed the reporting of statistics behind this observation. The acidification reports the physiological properties of the fungus; therefore, the reduced acidification in the presence of CoEvo2 and *degU*^{S202G} strains, that is now clearly presented and quantified, contribute to demonstrating the influence of CoEvo2 on the fungal growth.

4. When talking on the spreading of the evolved strains, which is the medium used in the experiments of interaction Bacillus-Aspergillus? It is said that CoEvo3 spread more in the presence of the fungus, but I missed how the colony looks in the same medium alone.

This data was included in the supplementary material. CoEvo3 clearly spreads less in the presence of the fungus. As indicated in the methods, all spreading experiments have been performed on LB medium.

5. It is really interesting the effect of the volatiles on the spreading of the colonies. However, I do feel that it is kind of superficial at this point, and that could be removed from the study without affecting to the main message. With all the information at genomic level, and the battery of mutants collected it could be explore which pathways are affected and whether they are somehow related to DegS-DegU. **We do not fully understand the exact suggestion which pathways should be examined. We agree that a transcriptome on CoEvo2 and the *degU*^{S202G} strains would be interesting to perform, but we feel that this is out of scope in the current study that explores the mutational landscape, reports the quantification of natural product surfactin, and explains the observed phenotypic changes by reintroducing the key mutations from CoEvo2 and CoEvo3. We have now moved all volatile experiments to supplementary material. That information is crucial to explain why CoEvo3 isolate is not spreading in the presence of the fungus, but able to spread more when cultivated without the fungus. However, we understand that this data might fragment the thread of the manuscripts message, therefore, we have moved this data fully to the supplementary information.**

6. It appears that the spreading phenotype of on the the coevolved strains is related to the overproduction of surfactin. Did the auhtor have found some realtion of surfactin overproduction and the mutation in the DEG system? or other wise these are complementary phenotypes. It would be compelling trying to find a putative connection between them. Given that clean mutants have been made on this loci, I consider relevant to analyze the level of production of surfactin and at least plipastatin in this builded strains, a finding that would help reinforcing the conclusion. Have the authors investigated how are the biofilm releated genes expressed in the evolved strain? Aghought it appears that the role of surfactin in triggering biofilm formation is strain dependent, adding some information would help understanding the spreading phenotype (directly related tos rufacitn, or indirectly).

We do not observe major influence on biofilm development in the co-culture evolved isolate carrying *degU*^{S202G}, as demonstrated using colony biofilms (Fig 1c and Supplementary figure 2d). Surfactin levels were tested using drop collapse assay using WT, Δ *srfAC*, CoEvo2 and *degU*^{S202G} strains. The data and its quantitative analysis with respective statistics is now included in the supplementary figure 4a. Previous publications reported that DegSU system influences expression of *bslA* gene (in addition to having influence on single cell motility), but not the other biofilm components encoding operons (DOI: 10.1111/j.1365-2958.2007.05923.x, 10.1128/JB.01170-10, and 10.1111/j.1365-2958.2007.05810.x).

7. In the study of the damage inflicted to Aspergillus, it is concluded that surfatin is the main responsible for the antagonistic interaction. First, it surprised me the absence of effect of plipastatin, however I do not really know if plipastatin is not antifungal or simply it is failing in the induction of budging. Second, in the luciferase experiments, I missed the effect of pure surfactin or plipastatin to support the findings at macroscopic or or microscopic level. The experiments where surfactin is shown to modify the fungal hyphae has been done in liquid or solid? What happened if surfactin is added in solid medium inoculated with Aspergillus? In this experiment we might find a similar effect with no interpherence related to bacterial cells.

All experiments related to the microscopy to visualize bulbous cells has been performed in liquid culture, as described in the methods section and now added to the figure legend for easy access to this information. Although we have previously tried, the deposition of surfactin on surfaces is challenging due to its amphiphilic properties. It is clearly demonstrated that in the absence of surfactin (the surfactin mutant still produce plipastatin, as previously published, see Kiewalter et al 2021 *mSystems* doi: 10.1128/mSystems.00770-20), the bacterial spent medium does not induce bulbous cells. Yes, the major influence of surfactin is surprising, nevertheless *Aspergillus* might be differently influenced by plipastatin, unlike *Fusarium* shown in the supplementary data. To connect the data observed microscopically and in the luciferase experiments, we refer to Supplementary figure 5 panels a and e, where it is clear that lack of surfactin in the bacterial supernatant results in lack of bulbous cells as well as reduced induction of cell wall stress. We cannot exclude that plipastatin has minor additional influence on inducing cell wall stress; however, in this current work, we aimed to demonstrate the major influence of surfactin on *A. niger*, which is supported by all experiments. This was motivated by increased surfactin production in the CoEvo isolates, on which we focus in this current study.

Third, how conserved is the pathway activated in *Aspergillus* in other fungi? If pretty well conserved, then one would expect a similar response in *Fusarium*, which is not.

The cell wall stress pathway in *Fusarium* similarly includes kinases and has been previously characterized (see for example DOI: 10.1105/tpc.110.075093 or DOI: 10.21769/BioProtoc.1915). Notably, we did not examine cell wall stress pathway in *Fusarium*, we simply report that bulbous cells are observed upon different cyclic lipopeptide. Further studies will be necessary (e.g. transcriptional, proteomics, analysis of the fungal membrane and cell wall compositions, use of a greater array of mutants both in *Aspergillus* and *Fusarium*) to depict the difference between the sensitivity of the two fungi towards lipopeptide. We believe that these experiments are outside of the scope of this current manuscript, that concentrates on the adaptation of *B. subtilis* to the presence of *A. niger* and the characterization of CoEvo2 isolates that produce increased level of surfactin.

8. Surfactin and other CLPs are known to target membranes but not the cell wall. Also, the authors have shown some experiments suggesting alterations of membranous systems but not cell wall. Thus I would not conclude on a direct action surfactin on the cell wall. That effect can be the conclusion of a sequence of events affecting the cell membrane and other citological disorders. I would like seen some ultrastructural studies of fungal cell, to really appreciate the alterations suggested with the fluorescent probes.

We apologize for the brevity of our text; we realize that some of our sentences might have been misleading; we do not propose that cell wall is targeted directly by surfactin. We propose that cell wall stress is induced that is clearly demonstrated by the higher reporter activity and the influence of RlmA in the process. Such cell wall stress could potentially originate from the reduced secretory vesicle delivery to the tip of the hyphae, as we now propose this hypothesis more clearly.

Is it surfactin killing *Aspergillus* cells, or it is only inducing this morphological changes? I think I did not see any experiments on viability or antifungal activity. Surfactin is not among the most potent antifungal CLPs produced by *Bacillus*.

Quantitative analysis of anti-fungal activity has been tested using the ancestor, CoEvo2, and *degU*^{S202G} strains, as we have previously developed (Kjeldgaard et al 2022 *Microbiol Spectrum* doi: 10.1128/spectrum.01433-21). The data are now included in the supplementary material and described in the material and methods that show reduced growth of *A. niger* when strains are used with enhanced surfactin production. Also, surfactin mutant *B. subtilis* clearly has reduced competitiveness against *A. niger*, as demonstrated by the increase medium acidification by *A. niger* when inoculated next to the *srfAC* mutant bacterium (see Supplementary figure S2 panels a and b).

Reviewer #1 (Remarks to the Author):

I appreciate the various responses the authors have provided and I think the overall clarity of the work is stronger. But the manuscript still needs considerable improvements. Some of the responses to reviewer comments are not fully developed and need more work.

For example, I noted how the intro was weak in the first version and lacked any kind of hypothesis about what might happen. In the revised version, the authors add a single sentence paragraph to address this that is fully underdeveloped. Why do you have these predictions?

The first paragraph is also not well structured or organized. I would break it into at least two paragraphs. What are the main themes you want to convey here to pull the reader into your paper? It drifts around quite a bit and you lose some main themes. I would start a new paragraph at the line "In addition to influencing short-term microbial.."

for "BFIs can also impact the evolution of an organism over longer time scales" you give one example, but there are many more in the literature. If you build this out more, you will be better able to add the hypotheses that I mention above.

As another review points out (and I hinted at with my notes about the intro being poorly written in the first version) there is very little background on bacterial evolution in the intro. This needs to be fixed so that the reader is primed to think about the research context for your work.

I appreciate that the authors agree that they lack an ecological context, but I would like to see them note that in some way in the Discussion. What does that limitation mean for readers interpreting your results?

I appreciate how the authors have cleaned up the overall structure of the manuscript by moving some things to the supp info

Reviewer #2 (Remarks to the Author):

I would like to compliment the authors on the work they have done, taking into account the comments made by the reviewers. The subject of the article is now much more focused and a great deal of additional data has been added. The new version thus forms a logical whole. Nonetheless, the new version still suffers from a formal point of view, in particular with a certain amount of key information missing from the text and/or the legends, which sometimes makes it difficult to read and interpret. See details below.

L119. « After 10 weekly transfer.. » It would be very useful here to mention that the two replicates are further called X.1.1, X1.2 (e.g. CoEvo1.1, CoEvo1.2) and to detail what does it mean when you talk about CoEvo1 vs CoEvo1.1/1.2, because it is very confusing to see the replicates popping up in some figures without explanations. Clarifying this point here in the main text and further in the figure legend would be very helpful in my mind.

Line 125. Fig1b. I think it would be better to replace figure 1b by Figure S1b to show all the phenotypes instead of only CoEvo2 as there are also a response of CoEvo4, and I think that it is a key result on which the full paper is based so it should not be hidden in the supplementary data.

Line 136-138 : limited vs restricted : I find the use of these verbs to describe the action of the strains imprecise and it took me a while looking at the picture to understand what you meant here, wondering if limited is stronger than restricted. Maybe replace « limited » by « almost suppressed » . In addition, I think that figure S2b is much more demonstrative than pictures and I would recommend to emphasize and describe results based on this quantitative graph rather than picture.

Regarding the legend of FigS2b, comparison of mean were made using ANOVA+ post-hoc test, which make sense but you need to explain for the stars which pairs are considered in the post-hoc test to be significant ; is only against the ancestor or all paired together ? If it is all paired together than you must put on the graphs a sign that indicate which pairs have different means.

Line 139 : srfAC mutant. This is the first time this mutant is mentioned, it would be worth helping the reader by describing which gene is targeted here and why. Otherwise the reader does not know how to interpret the result and how you reach the conclusion about the link between surface and pH.

Line 152, 156. Bac3.2 ; CoEvo4.1 ; This is the first time in the text that you mention the results obtained with the two CoEvo replicates; as I said earlier, I think it's very important that you specify why you sometimes use the data from the two replicates and why you sometimes summarise them. (e.g CoEvo4 instead of CoEvo4.1 and 4.2). It is actually striking to see that the two replicates can have very different phenotypes, which is not very surprising based your population genomics data but it raises questions about what data are used to produce the results in which you do not show the two replicate and how much we can trust them(e.g. Fig1b, Fig1c, Suppl Fig 1, 2...). I think this is a very important point to clarify in order to gain the reader's trust.

Fig2A : it would be interesting to show on that figure where the final coevolved strains are ; are they among the dominant genotypes or not ? What is depicted in grey ; it is not specified in the caption nor in the legend.

Fig2E : it would be useful to indicate in the legend in which genetic background are introduced the mutations. Results obtained with empty plasmids should be provided in supplemental data.

Suppl Fig3d : why did you use several starting concentrations for this experiment only? How different/similar are these concentrations with all the other experiments done ? If you use Tukey-Post hoc test, then you need to provide which pairs are statistically different.

Paragraph « Surfactin production is enhanced in CoEvo2

I think it would be worth starting this section by describing the different lipopeptides that the bacterial strain is able to produce. Indeed the question whether it can produce other lipopeptide that surfactin comes to mind when reading the results. The answer comes later but having it earlier would simplify the flow, in my mind.

How do you explain that CoEvo3 produces more surfactin than the ancestor but does not show much overgrowth nor inhibitory activity (in Figure S1b, Fig S2a) ?

Line 231 « the ability to produce non ribosomal proteins (plipastatin...) » Please rephrase ; there is a confusion between the fact that mutant cannot synthesise the NRPS proteins and the surfactants : NRPS do not code for plipastatin but are involved in their biosynthesis ; thus the mutation of the genes encoding for the NRPS blocks the synthesis of the surfactants

Line 238 – 241 : « In particular, secretory-vehicle specific... ». I find it difficult to reach this conclusion based on the pictures provided. Is the green signal in the cytoplasm aspecific signal ? What kind of imaging was done ; I did not find any information in the material and method section about the microscopy experiment ; is it epifluorescence, confocal, with lasers... please provide complete methods and update legends of the figures with proper information to allow interpretation of the images.

Suppl Fig4a. Stats are missing for CoEvo3

Fig4f : what are the two colonies spotted on the top part of the plates. This is the first time this appears, please describe.

Overall, regarding the spelling of mutant lines, there is an international nomenclature to spell/write mutant lines, that is used from time to time in the manuscript. Please use it all over the manuscript and be homogenous, it helps a lot the reader to follow what kind of mutant are being used (e.g. in the legend of Figure 4, there is a mix of ways to describe the mutants for instance).

Discussion section.

I think the verb is missing in the first sentence.

Overall, I am surprised that you do not discuss at all the results regarding population genomics

(Fig2) ; I think it's a strong and very interesting result of this experiment that would deserve to be highlighted and discussed.

Line 316 : « A. niger acidifies the medium by citric acid secretion ». If so, why did you use mutant that do not produce oxalic acid in your first experiments, and why they do not acidify anymore the medium ? Please check and correct as appropriate.

Reviewer #3 (Remarks to the Author):

In this new version of the manuscript the authors have provided with satisfactory answers to all my questions and suggestions. The manuscript has clearly benefited from the revision. It is indeed a beautiful piece of work.

Reviewer #1 (Remarks to the Author):

I appreciate the various responses the authors have provided and I think the overall clarity of the work is stronger. But the manuscript still needs considerable improvements. Some of the responses to reviewer comments are not fully developed and need more work.

We have now followed the precise comments raised by Reviewer 1 during the revision stage that helped us to improve the manuscript.

For example, I noted how the intro was weak in the first version and lacked any kind of hypothesis about what might happen. In the revised version, the authors add a single sentence paragraph to address this that is fully underdeveloped. Why do have these predictions?

We have now explained the few available publications testing the experimental evolution of a bacteria-fungi interaction, which motivates our study. There is an extensive literature on phenotypic adaptation during BFI, but less is available on the evolution of these interaction, hence our work is novel in the field.

The first paragraph is also not well structured or organized. I would break it into at least two paragraphs. What are the main themes you want to convey here to pull the reader into your paper? It drifts around quite a bit and you lose some main themes. I would start a new paragraph at the line "In addition to influencing short-term microbial.."

We have introduced the new paragraph as suggested and decided to shorten down the various examples of phenotypic adaptation. We agree with the reviewer that the evolution of BFI is the main message, thus we explain few previous examples on this area.

for "BFIs can also impact the evolution of an organism over longer time scales" you give one example, but there are many more in the literature. If you build this out more, you will be better able to add the hypotheses that I mention above.

We have included additional examples, and highlighted examples on the experimental evolution of BFI.

As another review points out (and I hinted at with my notes about the intro being poorly written in the first version) there is very little background on bacterial evolution in the intro. This needs to be fixed so that the reader is primed to think about the research context for your work.

*We have extended the bacterial evolution part, especially those known for *B. subtilis*.*

I appreciate that the authors agree that they lack an ecological context, but I would like to see them note that in some way in the Discussion. What does that limitation mean for readers interpreting your results?

We have indicated the limitations of our study on the ecology side, highlighted the natural co-occurrence of B. subtilis with fungi and the variation in surfactin production by natural isolates, and finally indicated the future study will be necessary to reveal similar genetic changes observed in our experiments. This fits nicely before the final conclusion that our methodology can be used for industrial improvement of Bacilli against fungi without the need for GMO.

I appreciate how the authors have cleaned up the overall structure of the manuscript by moving some things to the supp info.

We are glad to hear this approach is acknowledged.

Reviewer #2 (Remarks to the Author):

I would like to compliment the authors on the work they have done, taking into account the comments made by the reviewers. The subject of the article is now much more focused and a great deal of additional data has been added. The new version thus forms a logical whole. Nonetheless, the new version still suffers from a formal point of view, in particular with a certain amount of key information missing from the text and/or the legends, which sometimes makes it difficult to read and interpret. See details below.

We thank Reviewer 2 for providing specific comments on the revised manuscript that allows us to answer these aspects.

L119. « After 10 weekly transfer... » It would be very useful here to mention that the two replicates are further called X.1.1, X1.2 (e.g. CoEvo1.1, CoEvo1.2) and to detail what does it mean when you talk about CoEvo1 vs CoEvo1.1/1.2, because it is very confusing to see the replicates popping up in some figures without explanations. Clarifying this point here in the main text and further in the figure legend would be very helpful in my mind.

Thank you for this suggestion, we have now indicate that isolate 1 from each lineage was used, unless indicated specifically in the text. This is indeed crucial information.

Line 125. Fig1b. I think it would be better to replace figure 1b by Figure S1b to show all the phenotypes instead of only CoEvo2 as there are also a response of CoEvo4, and I think that it is a key result on which the full paper is based so it should not be hidden in the supplementary data.

During the first round of reviews, we were advised to concentrate on the main isolate, CoEvo2, however, we agree that including all panels in the main figure is useful. Therefore, we followed these new suggestion and transferred all panels from Fig S1b to Fig 1.

Line 136-138 : limited vs restricted : I find the use of these verbs to describe the action of the strains imprecise and it took me a while looking at the picture to understand what you meant here, wondering if limited is stronger than restricted. Maybe replace « limited » by « almost suppressed » . In addition, I think that figure S2b is much more demonstrative than pictures and I would recommend to emphasize and describe results based on this quantitative graph rather than picture.

This is a great suggestion, we have adjusted the text as suggested and emphasized the quantitative measurement of the acidified area in the text.

Regarding the legend of FigS2b, comparison of mean were made using ANOVA+ post-hoc test, which make sense but you need to explain for the stars which pairs are considered in the post-hoc test to be significant ; is only against the ancestor or all paired together ? If it is all paired together than you must put on the graphs a sign that indicate which pairs have different means.

This is relevant, we have indicated the comparison to the ancestor in the legend.

Line 139 : srfAC mutant. This is the first time this mutant is mentioned, it would be worth helping the reader by describing which gene is targeted here and why. Otherwise the reader does not know how to interpret the result and how you reach the conclusion about the link between surface and pH.

Thank you, indeed, this is important information, it is now indicated in the sentence.

Line 152, 156. Bac3.2 ; CoEvo4.1 ; This is the first time in the text that you mention the results obtained with the two CoEvo replicates; as I said earlier, I think it's very important that you specify why you sometimes use the data from the two replicates and why you sometimes summarise them. (e.g CoEvo4 instead of CoEvo4.1 and 4.2). It is actually striking to see that the two replicates can have very different phenotypes, which is not very surprising based your population genomics data but it raises questions about what data are used to produce the results in which you do not show the two replicate and how much we can trust them(e.g. Fig1b, Fig1c, Suppl Fig 1, 2...). I think this is a very important point to clarify in order to gain the reader's trust.

This is now clearly indicated in the text based on the above comment. This is indeed important information to be described.

Fig2A : it would be interesting to show on that figure where the final coevolved strains are ; are they among the dominant genotypes or not ? What is depicted in grey ; it is not specified in the caption nor in the legend.

Based on the specifically discussed mutations, e.g. degU and degS SNPs, are indeed dominant genotypes represented 100% of the populations (see Supplementary Dataset S1). This is now indicated in the manuscript when discussing the population sequencing dataset, as requested below). Grey color includes nested genotypes, which is now added indicated in the legend.

Fig2E : it would be useful to indicate in the legend in which genetic background are introduced the mutations. Results obtained with empty plasmids should be provided in supplemental data.

The specific mutations (either deletion or single nucleotide exchange) have been introduced to the genome of the ancestor, thus no empty plasmid is available. The strains are clearly described in the materials and methods and corresponding strain table. However, we agree with the Reviewer that this information is useful, so it is now indicated in the figure legend.

Suppl Fig3d : why did you use several starting concentrations for this experiment only? How different/similar are these concentrations with all the other experiments done ? If you use Tukey-Post hoc test, then you need to provide which pairs are statistically different.

We used different starting concentrations motivated by our experimental setup developed by Kjeldgaard et al 2012 Microbiology Spectrum, that also serves to demonstrate robustness of the assay over broader range of starting OD. This has been added now to the legend. The middle panel with OD_{600} of 0.004 corresponds closely with other assays when bacterial cells have been added to fungal cultures. In the legend, we have now indicated the comparison to ancestor.

Paragraph « Surfactin production is enhanced in CoEvo2

I think it would be worth starting this section by describing the different lipopeptides that the bacterial strain is able to produce. Indeed the question whether it can produce other lipopeptide that surfactin comes to mind when reading the results. The answer comes later but having it earlier would simplify the flow, in my mind.

The B. subtilis lipopeptides are now introduced at the start of the paragraph, as requested by the reviewer. Good suggestion.

How do you explain that CoEvo3 produces more surfactin than the ancestor but does not show much overgrowth nor inhibitory activity (in Figure S1b, Fig S2a) ?

This is clearly explained by the A. niger produced volatile compounds that is included in the supplementary results, that was requested by Review 1 and 3 during the first review round to be removed from the main body of the results. Only one sentence remained in the results section explaining these observations: "Preliminary experiments

suggest that fungal produced volatile compounds, dimethyl disulphide, pyrazine, and 1-pentyne restrict CoEvo3 spreading (Supplementary results 1).”

Line 231 « the ability to produce non ribosomal proteins (plipastatin...) » Please rephrase ; there is a confusion between the fact that mutant cannot synthetise the NRPS proteins and the surfactants : NRPS do not code for plispatatin but are involved in their biosynthesis ; thus the mutation of the genes encoding for the NRPS blocks the synthesis of the surfactants

Correct, the sentence has been corrected to “other non-ribosomal proteins (those that are involved in the synthesis of plipastatin and bacillaene)”

Line 238 – 241 : « In particular, secretory-vehicle specific... ». I find it difficult to reach this conclusion based on the pictures provided. Is the green signal in the cytoplasm aspecific signal ? What kind of imaging was done ; I did not find any information in the material and method section about the microscopy experiment ; is it epifluorescence, confocal, wich lasers... please provide complete methods and update legends of the figures with proper information to allow interpretation of the images.

Thank you for highlighting the missing material and methods on CLSM that was used to visualize the fluorescent proteins localized to the secretory vesicles. This has been now added. The non-treated fungal hyphae contains secretory vesicles majorly localized to the tip of the fungal hyphae as reported previously by Kwon et al (cited as reference at the end of the sentence). In the surfactin treated hyphae, as we describe n the results, these vesicles are not localizaed to the tip of the hyphae and fully mislocalized in the bulbous cells. This is not autofluorescence, as thereshold during imaging was set to omit autofluorescence (this now indicated in the materials and methods) and non labeled Aspergillus show no fluorescence signal. We do not know the reason behind full dispersal of fluorescence signal in the bulbous cells, this will require further experiment to determine if GFP is cleaved from the SncA protein or the SncA protein is dispersed in the cells. This is now explained in the corresponding results.

(a) —

PsncA	AopyrG	PsncA	gfp	sncA	TsncA
-------	--------	-------	-----	------	-------

 —

Figure 1 from Kwon et al 2014

Suppl Fig4a. Stats are missing for CoEvo3

The p value is 0,0500435, therefore it is not indicated as a significant value.

Fig4f : what are the two colonies spotted on the top part of the plates. This is the first time this appears, please describe.

This has been now explained in the figure legend. Thank you for this suggestion.

Overall, regarding the spelling of mutant lines, there is an international nomenclature to spell/write mutant lines, that is used from time to time in the manuscript. Please use it all over the manuscript and be homogenous, it helps a lot the reader to follow what kind of mutants are being used (e.g. in the legend of Figure 4, there is a mix of ways to describe the mutants for instance).

Thank you for noticing that sfp mutant was not italicized. This is now corrected. For the Aspergillus mutant, we prefer to use the nomenclature used in the original publications that describe the specific mutant strains, thus strains with protein fusions are symbolized with proteins instead of genes in the introduced constructs. Anyhowm these are not mutants, but fusions introduced into the fungus.

Discussion section.

I think the verb is missing in the first sentence.

Thank you, now corrected.

Overall, I am surprised that you do not discuss at all the results regarding population genomics (Fig2) ; I think it's a strong and very interesting result of this experiment that would deserve to be highlighted and discussed.

We have extended the discussion of the population genomics data in the manuscript as well as on the specific mutations.

Line 316 : « A. niger acidifies the medium by citric acid secretion ». If so, why did you use mutant that do not produce oxalic acid in your first experiments, and why they do not acidify anymore the medium ? Please check and correct as appropriate.

Thank you for catching this mistake, it should have been organic acid, including oxalic acid. The sentence has been corrected.

Reviewer #3 (Remarks to the Author):

In this new version of the manuscript the authors have provided with satisfactory answers to all my questions and suggestions. The manuscript has clearly benefited from the revision. It is indeed a beautiful piece of work.

We are grateful for the original comments of Reviewer 3 and acknowledging the revisions.